# Riboformer: a deep learning framework for predicting context-dependent translation dynamics

Bin Shao ⬚[1,6] ✉, Jiawei Yan[2], Jing Zhang ⬚[3], Lili Liu[4], Ye Chen[4] & Allen R. Buskirk ⬚[5]

Translation elongation is essential for maintaining cellular proteostasis, and alterations in the translational landscape are associated with a range of diseases. Ribosome profiling allows detailed measurements of translation at the genome scale. However, it remains unclear how to disentangle biological variations from technical artifacts in these data and identify sequence determinants of translation dysregulation. Here we present Riboformer, a deep learning-based framework for modeling context-dependent changes in translation dynamics. Riboformer leverages the transformer architecture to accurately predict ribosome densities at codon resolution. When trained on an unbiased dataset, Riboformer corrects experimental artifacts in previously unseen datasets, which reveals subtle differences in synonymous codon translation and uncovers a bottleneck in translation elongation. Further, we show that Riboformer can be combined with in silico mutagenesis to identify sequence motifs that contribute to ribosome stalling across various biological contexts, including aging and viral infection. Our tool offers a context-aware and interpretable approach for standardizing ribosome profiling datasets and elucidating the regulatory basis of translation kinetics.

Ribosomes move along mRNAs at varying rates, which can impact protein homeostasis and cellular function[1–3]. Elongation rates across the transcriptome are shaped by a complex interplay between local sequence features, such as mRNA secondary structures[4], clusters of charged amino acids[5], and consecutive proline residues[6], and global factors like cellular resource availability and protein quality control[7–9]. These intricacies impact translation efficiency, co-translational protein folding, and covalent modification[1,3,10–12]. Cells must fine-tune elongation rates to achieve the proper levels of protein output from each mRNA, accounting for demands of regulation and protein folding. Despite recent advances in understanding translation dynamics, deciphering the regulatory code of translation

dysregulation and proteostasis collapse in complex diseases remains challenging[13,14].

The advent of ribosome profiling has led to substantial progress in our understanding of mRNA translation[8]. Ribosome profiling captures and sequences mRNA fragments protected by ribosomes from nuclease digestion, allowing the reliable inference of the ribosomal decoding site in each footprint and yielding information about ribosome distribution along mRNA from each gene. In general, the more ribosome density there is on a codon, the slower it is decoded. With improved methods, non-optimal codons were found to have higher ribosome density and be decoded more slowly, as expected[15,16]. Several computational approaches have been developed to glean insights

[1]Department of Molecular and Cellular Biology, Harvard University, Cambridge, MA, USA. [2]Department of Chemistry, Stanford University, Stanford, CA, USA. [3]Biological Design Center, Boston University, Boston, MA, USA. [4]Key Laboratory of Quantitative Synthetic Biology, Shenzhen Institute of Synthetic Biology, Shenzhen Institute of Advanced Technology, Chinese Academy of Sciences, Shenzhen, China. [5]Department of Molecular Biology and Genetics, Johns Hopkins University School of Medicine, Baltimore, MD, USA. [6]Present address: Klarman Cell Observatory, Broad Institute of MIT and Harvard, Cambridge, MA, USA. ✉e-mail: shaobinlx@gmail.com

from the accumulating body of ribosome profiling data that are publicly available. Whole-cell models based on these data provide a precise depiction of the physical process of translation[17,18]. Sophisticated models, such as probabilistic models and neural network models, have been used to study how ribosome density is determined by the mRNA sequence and biophysical features of the nascent polypeptide[7,9,19–22]. For example, Ribo-seq Unit Step Transformation (RUST) identifies positional mRNA sequence features that affect ribosome footprint densities and predicts ribosome density with high accuracy[7]. A convolutional neural network (CNN) model has been implemented to predict ribosome stalling sites in both yeast and human cells[20], outcompeting the conventional methods. More recently, deep learning methods such as RiboMIMO[23] and Riboexp[24] were developed to reconstruct the ribosome density distributions based on the coding sequence (CDS).

Despite these computational advances, little effort has been devoted to model the context-dependent changes in translation dynamics. Consequently, it remains a challenge to distinguish biological signals from technical artifacts that have a profound effect on the observed translational landscape[7]. For example, Mohammad et al. found that methods used to arrest translation and harvest bacterial cultures introduce sequence-specific ribosome pauses[16]. Unfortunately, existing computational tools lack the ability to use multiple datasets (biased vs unbiased) to model the shift in ribosome distributions induced by these artifacts. Secondly, the underlying mechanism driving the changes in the translational landscape under complex physiological states remains largely elusive. Although disease-focused studies often employ design principles such as case versus control, current methods don't harness these approaches to uncover the sequence features that affect translation elongation in disease progression. Lastly, the predictive power of current models is limited. The trained models cannot be utilized to improve the analysis of existing experiments or predict ribosome distribution in new contexts.

To address these challenges, we present Riboformer, a deep learning-based framework that models the context-dependent changes in ribosome dynamics at codon resolution. Our model compares ribosome distributions between two datasets and extracts the sequence features driving the difference between them. This structure enables the trained Riboformer model to remove experimental bias from the input dataset, query the sequence determinants of relative changes in ribosome density, and predict sites of ribosome collision (disome) from monosome profiles (Fig. 1a). Our approach uses a transformer architecture that effectively captures interdependencies between codons in the regulation of translation elongation[25] (Fig. 1b). We have benchmarked the prediction performance of Riboformer using a variety of prokaryotic and eukaryotic ribosome profiling datasets. We demonstrate the effectiveness of our neural network structure in modeling the impact of experimental protocols on the in vivo translational landscape, and the trained Riboformer model corrects artifacts in a wide range of unseen datasets. This process reveals subtle differences in synonymous codon translation and uncovers a potential bottleneck in translation elongation. Combined with in silico mutagenesis analysis, Riboformer identifies peptide motifs that contribute to ribosome stalling across various biological contexts, such as aging and viral infection, highlighting its versatility in diverse research areas (Fig. 1a). Altogether, Riboformer is an end-to-end tool that facilitates the standardization and interpretation of ribosome profiling datasets, and our results demonstrate the potential of context-aware deep learning models that capture the complex dynamics of biological processes subject to variations in cell physiological states. Riboformer is implemented in Python as a command line tool[26], publicly available at https://github.com/lingxusb/Riboformer/

## Results

### Riboformer accurately clarifies ribosome density

Training of Riboformer requires two ribosome profiling datasets and it leverages a transformer architecture to capture the sequence features that determine the changes in translation kinetics (Fig. 1b). The transformer block consists of self-attention layers that gather the impact of distant codons based on their sequence representations[25], in contrast to convolutional neural network that relies on convolution operators to detect local sequence motifs. The first input to our model is the reference dataset consisting of normalized ribosome density from a control experiment as a baseline for modeling translation dynamics. The second input to our model is the coding sequence. More specifically, our approach assumes that the relative change in ribosome occupancy between the reference dataset and the target dataset is primarily determined by the surrounding sequences. The codon sequence around the position of interest and the normalized ribosome footprint counts in the control experiment were encoded as vectors, which were further connected to two branches of neural networks. The features extracted from the two inputs by a series of transformer blocks were subsequently merged using element-wise multiplication. Finally, a fully connected layer converts the output to the normalized ribosome density in the target condition (see the "Methods"). Thus, Riboformer learns how to convert ribosome density from one condition to another based on the differences it observes in the training datasets.

To evaluate the performance of Riboformer, we started with bacterial samples in which technical artifacts during the preparation of the libraries had perturbed the underlying translation kinetics. Historically, bacterial samples were commonly harvested by rapid filtering and lysed in a buffer containing chloramphenicol (Cm) to arrest elongation[27]. However, recent ribosome profiling and toeprinting studies have found that this protocol alters translation elongation in a sequence-specific manner[16,28]. To address this issue, a novel protocol was developed that involves flash-freezing the cell culture directly and arresting translation with a lysis buffer containing high magnesium concentrations[16]. This approach eliminates pauses at Ser and Gly codons arising from the filtering protocol and provides a clearer view of the in vivo translational landscape. We trained our Riboformer model on this dataset to predict the unperturbed ribosome profile (Mg) based on the perturbed profile (Cm). The input sequence included instances of the codon of interest across all expressed genes (methods) as well as the sequence and ribosome density of 20 codons upstream and downstream. The normalized ribosome densities from the two experiments were used as the inputs (Fig. 2a). More specifically, we chose the 1005 genes with the highest ribosome densities to construct a dataset of 323,688 instances of codons. Then we used 10-fold cross-validation tests to evaluate the model performance. In each fold, one-tenth of the data was held out as test data while the remaining data were used as training data. We used Pearson and Spearman correlation coefficients to measure the correlation between the predicted and true ribosome densities for all codons in the test datasets.

As shown in Fig. 2b, starting with samples obtained by filtering with the Cm-lysis buffer, Riboformer accurately predicts the codon-level ribosome density of samples obtained by flash-freezing with the high-Mg buffer. There is a high correlation between the ground truth and the predicted ribosome density ($r = 0.91$, Fig. 2b, Supplementary Table 1). We defined the ratio of ribosome occupancy at each codon to the average ribosome occupancy of the CDS as the codon pause score, and we found that Riboformer recapitulated the average pause score for all the codons (Fig. 2c, see the "Methods" section). Notably, ribosome pausing at Gly and Ser codons is largely reduced, and Pro has a high pause score at all three ribosomal tRNA binding sites (E, P, A) in the corrected profiles[16].

We further investigated how the input data characteristics affect the model performance. By varying the window sizes of the input

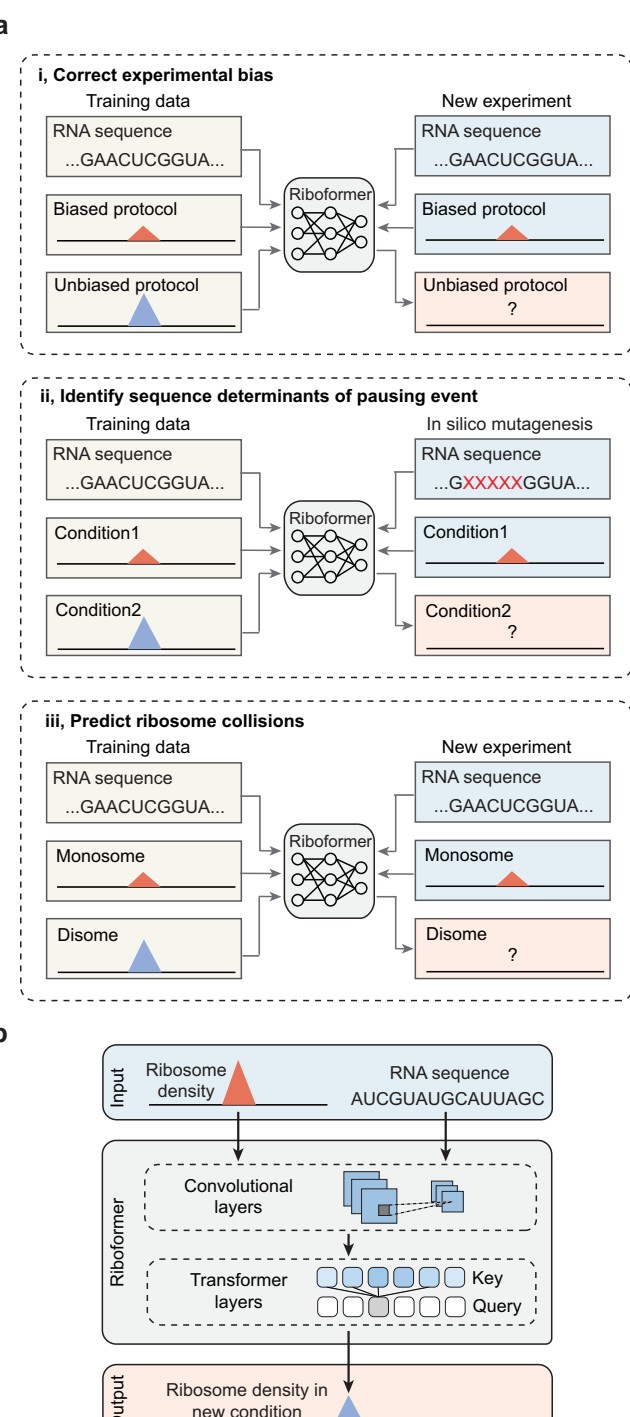

**a**

**i, Correct experimental bias**

Training data | New experiment

RNA sequence ...GAACUCGGUA...

Biased protocol

Unbiased protocol

Riboformer

RNA sequence ...GAACUCGGUA...

Biased protocol

Unbiased protocol ?

**ii, Identify sequence determinants of pausing event**

Training data | In silico mutagenesis

RNA sequence ...GAACUCGGUA...

Condition1

Condition2

Riboformer

RNA sequence ...GXXXXXGGUA...

Condition1

Condition2 ?

**iii, Predict ribosome collisions**

Training data | New experiment

RNA sequence ...GAACUCGGUA...

Monosome

Disome

Riboformer

RNA sequence ...GAACUCGGUA...

Monosome

Disome ?

**b**

Input: Ribosome density / RNA sequence AUCGUAUGCAUUAGC

Riboformer: Convolutional layers / Transformer layers — Key, Query

Output: Ribosome density in new condition

**Fig. 1 | Overview of the Riboformer pipeline. a** Applications of the Riboformer pipeline. **b** Schematic illustration of the neural network structure of Riboformer.

accuracy is robust across replicated experiments and that increasing the number of replicates further enhances the model accuracy (Supplementary Table 3 and Supplementary Note 1). Finally, we systematically compared the performance of Riboformer with other deep learning-based models including RiboMIMO and Riboexp and found that it compares favorably (Supplementary Fig. 3, Supplementary Table 4, and Supplementary Note 2). In conclusion, our results demonstrate the robust performance of Riboformer across different input window sizes, gene expression levels, and replicated experiments.

## Riboformer corrects experimental bias in unseen data

We further used the trained *E. coli* Riboformer model to correct for the bias in the translational landscapes in other datasets produced with the same experimental artifacts. We applied it to an unseen ribosome profiling dataset obtained with filtering and the Cm lysis buffer (Supplementary Fig. 4) from *E. coli* cells with low levels of $m^1G37$ in tRNAs, a deficiency that affects the decoding of specific codons[29]. Using the trained Riboformer model to predict the unperturbed ribosome occupancy, we were able to correct bias in the pause scores for Gly codons, while maintaining the high pause scores for the affected Pro and Arg codons CCA, CCG, and CGG (Supplementary Fig. 4). Working with a second dataset prepared in a different lab[30], Riboformer removed the strong pauses at Ser and Gly codons and highlighted increased ribosome occupancy at Pro and Trp codons (Fig. 2e). Moreover, in a sample from this dataset overexpressing a transgene containing the rare Leu codon CUA, we observed a high pause score for the CUA codon in the corrected ribosome profiles due to increased demand for the corresponding tRNA, similar to the uncorrected results[30] (Fig. 2d and e). Together, these results show that the subtle variation of ribosome pausing in synonymous codons is preserved even as the experimental bias is removed. In addition, the ribosome occupancy from these samples was previously shown to be correlated with the level of genome-wide RNA structures determined by dimethyl sulfate (DMS)-seq[31]. Our corrected ribosome occupancy shows a higher correlation with the DMS-seq score (Supplementary Fig. 5) than originally reported[30], confirming the impact of mRNA secondary structure on translational efficiency[32]. Collectively, these results demonstrate that once trained on unbiased datasets, the Riboformer model can be used to standardize a wide range of ribosome profiling measurements, reducing experimental noise while remaining true to the underlying biological signal of interest.

## Riboformer identifies a limiting step in translation elongation

In synthetic biology, the proper functioning of engineered systems relies on the coordinated expression of functional genes. However, the expression of heterologous genes imposes an additional burden on the cells, which negatively impacts the growth rate and leads to evolutionary instability. Ribosome profiling has been used to quantify the consumption of cellular resources by a 3-input genetic circuit consisting of seven NOT/NOR gates in *E. coli* cells[33] (Fig. 3a). However, this dataset was generated using a biased protocol. To gain a better understanding of the translation dynamics in burdened cells, we used the trained Riboformer model to predict the unperturbed ribosome occupancy across the transcriptome in eight circuit states. We found a reduction in the pause scores of Glu, Ser, and Thr codons in the ribosomal A site, while Pro and Trp showed the highest pause scores (Fig. 3b). We then explored the relationship between translational efficiency (TE) of genes and codon pause scores. Translational efficiency was defined as the ribosome density (RD) normalized by mRNA level as quantified by RNA-seq. Interestingly, we found that genes with high TE tend to have a higher pause score for Trp (Fig. 3c, methods). Thus, our results indicate that slow decoding of Trp codons could affect translation elongation, potentially serving as a rate-limiting step in protein synthesis. To further characterize the role of pausing at Trp

sequence, we observed that the model performance increases with window size (Supplementary Fig. 1). However, the improvement becomes marginal when the window size exceeds 40 codons. We found the model performs better for highly expressed genes due to the high signal-to-noise ratio (Supplementary Fig. 2). Thus, we assessed potential biases in model performance that might arise when the model is trained on genes with high ribosome density. Interestingly, models trained on these genes could effectively predict ribosome density for more lowly expressed genes (Supplementary Table 2 and Supplementary Note 1). We also found that Riboformer's prediction

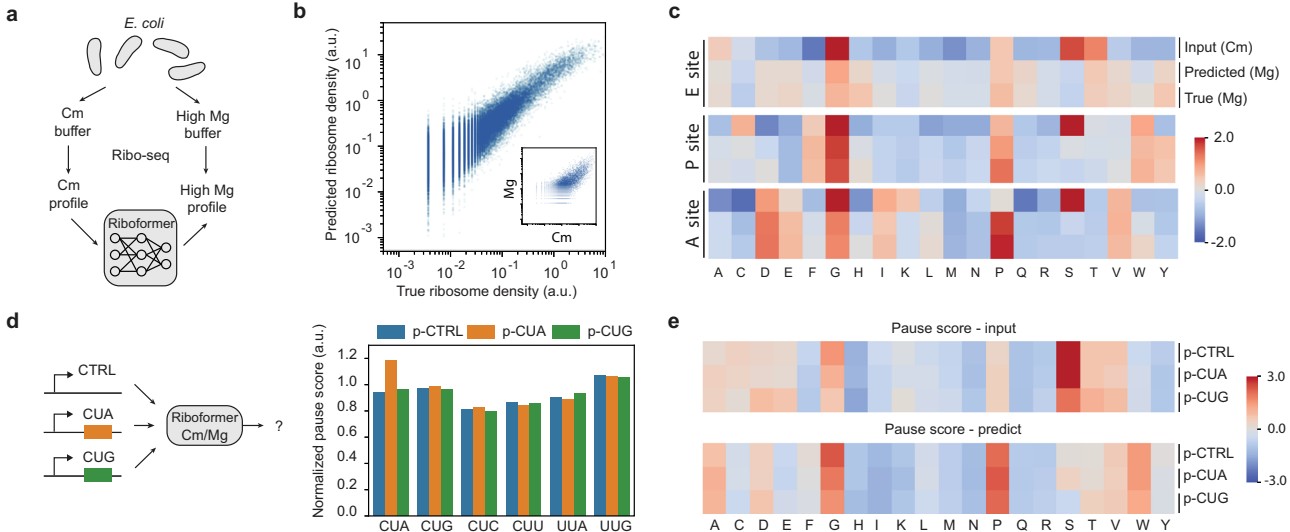

**Fig. 2 | Riboformer captures context-dependency of translation dynamics. a** *E. coli* cells were treated with different lysis buffers and the resulting profiling data were used to train the neural network model. **b** Correlation between the true and predicted ribosome density for all codon positions with high Mg buffer (Mg) in the test dataset (Pearson correlation coefficient, *r* = 0.91). Inset, ribosome density for all codons with high Mg buffer and Cm buffer (*r* = 0.75). **c** Heatmap of pause scores for all 20 amino acids. **d** Average ribosome occupancy at leucine codons in

endogenous genes after correction of experimental bias. The *E. coli* cells over-expressed a control plasmid (p-CTRL without a mini ORF) or plasmids with a het-erologous CUA mini-ORF (p-CUA) or a CUG mini-ORF (p-CUG). **e** mean codon pause scores for *E. coli* cells overexpressing mini-ORFs. Codon pause scores before (input) and after (predicted) experimental bias correction are shown. Source data are provided as a Source Data file.

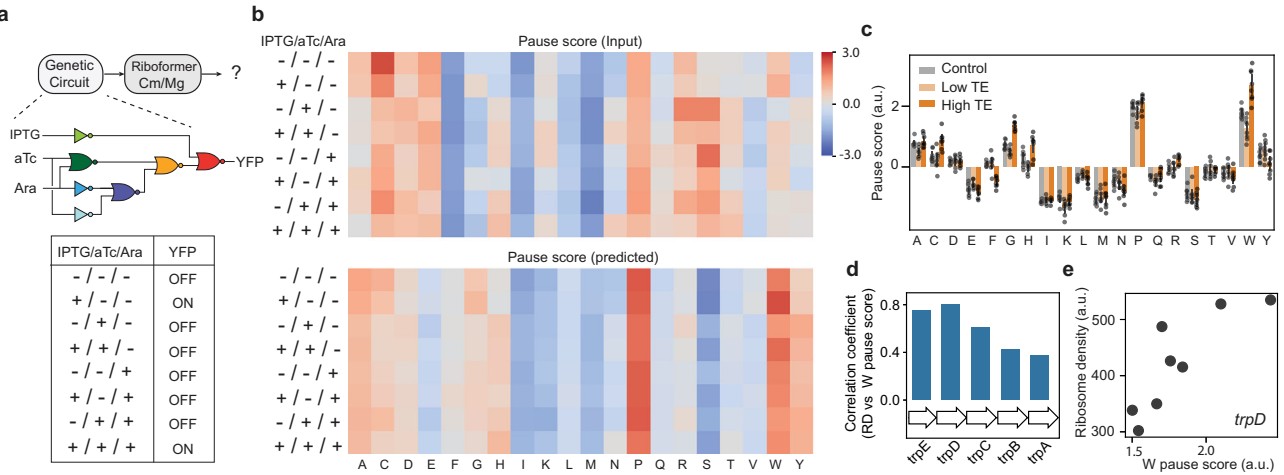

**Fig. 3 | Riboformer identifies a bottleneck in translation elongation. a** Overview of the procedure to remove experimental artifacts in ribosome profiles from cells with a large-scale synthetic circuit. The Riboformer model was trained on unbiased data (High-Mg buffer) and applied to the ribosome profiles generated using the Cm-lysis buffer. Each gate is associated with a specific transcriptional repressor, which is indicated by the corresponding color. **b** Mean codon pause scores for all

eight induction states before and after correction for experiment bias. **c** Pause score of 20 amino acids in genes with different translation efficiency (TE). Error bar represents SEM from eight induction conditions. **d** Correlation of mean Trp pause score with the mean ribosome density (RD) of the *trp* operon genes in eight induction states. **e** *trpD* gene expression (RD) and mean Trp codon pause scores in eight induction states (*r* = 0.80). Source data are provided as a Source Data file.

codons on gene expression in the strains expressing the engineered circuits, we calculated the correlation of the Trp pause score and the level of expression of the Trp biosynthesis genes for different circuit states, as quantified by the ribosome density (Fig. 3d). There was a positive correlation between the expression of Trp operon genes and the Trp pause score in the corrected ribosome profiles, especially for *TrpE* and *TrpD* (Fig. 3e). This observation is in accord with the well-characterized regulation of these genes by transcriptional attenuation after *trpL* which is upstream of *trpE*[34]. Ribosome stalling in the Trp codon-rich *trpL* sequence promotes transcription of the TrpEDCBA operon. The clarity in the pausing landscape provided by Riboformer

allows us to explain these changes in gene expression driven by overexpression of the circuit components in this example.

## Riboformer identifies sequence determinants of ribosome collisions

Prolonged slowing of translating ribosomes can lead to ribosome collisions, triggering ribosome rescue pathways that promote the degradation of the nascent polypeptide[35–37]. Collided ribosomes form nuclease-resistant disomes because they protect the mRNA at the disome interface. Disome profiling experiments allow the genome-wide detection of collided ribosomes by sequencing the disome-

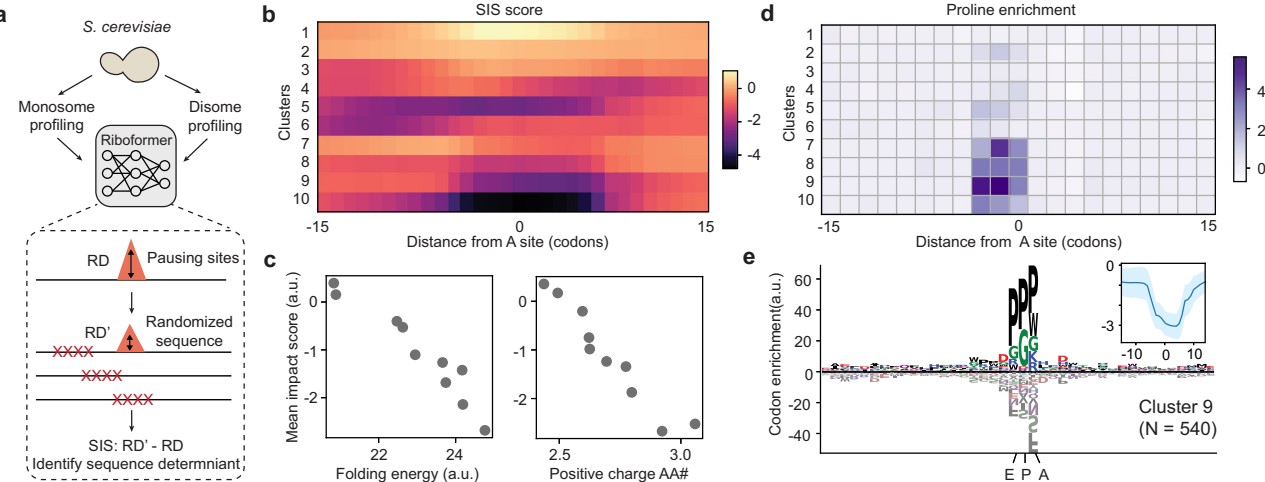

**Fig. 4 | Riboformer identifies sequence determinants of ribosome pausing.**
**a** Calculation of the sequence impact score (SIS). After the Riboformer model is trained, a moving window of the codon sequence is randomly mutated and the corresponding change in the predicted ribosome density (dRD) is recorded as SIS. **b** SIS profiles of all the ribosome stalling sites are grouped into 10 clusters. **c** mean SIS of each cluster versus the mean folding energy of the input mRNA sequence (left, $r = -0.96$), and the mean number of positive charged amino acids of the input sequence (right, $r = -0.92$). **d** Pro codon enrichment scores for all the clusters (methods). **e** Codon enrichment profile for cluster 9 (number of ribosome stalling sites: $n = 540$). Inset, mean SIS profile for cluster 9. The shaded region represents the standard derivation of all profiles. Source data are provided as a Source Data file.

protected mRNA fragments[38-41]. To examine the relationship between ribosome collisions and mRNA sequence features, we used Riboformer to identify the sequence determinants of ribosome collisions in budding yeast (*Saccharomyces cerevisiae*, Fig. 4a). Although the monosome and disome densities show a weak correlation across the genome (Supplementary Fig. 6a, $r = 0.35$), our framework successfully predicts the disome profiles based on monosome occupancy (Supplementary Fig. 6a, $r = 0.75$). For all sites with significant ribosome collisions ($n = 11,079$, Supplementary Fig. 6b), we used an *in-silico* mutagenesis approach to determine the sequences that contribute to ribosome stalling (Fig. 4a). In brief, a sliding window of the codon sequence was randomly mutated, and the corresponding change in the predicted disome occupancy at the position of interest was defined as the sequence impact score (SIS) for the mutation window (see the "Methods" section).

We performed unsupervised clustering of the SIS profiles for all the significant ribosome collision sites (Fig. 4b, see the "Methods" section) which were grouped into 10 clusters. Interestingly, we found that the mean SIS of each cluster is linearly correlated with the mean mRNA folding energy ($r = -0.96$, Fig. 4c). In other words, when the mRNA is highly structured, disrupting the mRNA sequence by introducing mutations leads to lower predicted levels of ribosome collisions, in agreement with a previous report[42]. In addition, positively charged amino acids have been shown to slow down ribosome elongation speed by interacting with the exit channel[9,43]. Our approach also identifies a strong negative correlation between the number of positive charged amino acids in the upstream sequence with SIS ($r = -0.96$, Fig. 4c), suggesting that removing these amino acids would reduce ribosome stalling. In contrast, the number of negatively charged amino acids shows little correlation with SIS ($r = -0.07$, Supplementary Fig. 6c).

Notably, a few clusters have their lowest SIS at the ribosome decoding sites (Fig. 4b, clusters 7–10), indicating that these ribosome collisions are mediated by local sequence features. In these clusters, Pro codons are enriched in all three tRNA binding sites (E, P, and A) (Fig. 4d and e), consistent with the well-characterized tendency of Pro residues to slow down translation elongation[44-46]. Clusters 4 and 5 show distinct SIS profiles from the population average, with Trp and Lys codons enriched at the ribosomal A site, respectively (Supplementary Fig. 7). Interestingly, the R-X-K motif of cluster 5 is enriched in

ribosome collision sites in both humans and zebrafish[41], and it aligns with the amino acid motifs associated with macrolide-induced ribosome arrest[47]. We found pausing sites from cluster 4 are more likely to be affected by the mRNA secondary structure of the downstream sequences (Supplementary Fig. 7c). In addition, previous works have demonstrated that consecutive Lys codons (polybasic region) could be potential sites for ribosome collision[13,40]. Riboformer further identifies consecutive Lys codons as the sequence determinant of disome peaks in the *PWP1* gene (Supplementary Fig. 8). In summary, our interpretable framework identifies the sequences responsible for ribosome collision events, clusters these sequences into distinct groups, and uncovers various modes of ribosome stalling, offering insights beyond motif analysis of all the pausing sites.

We further used the trained Riboformer model to identify novel disome sites in yeast from published monosome data[5] (Fig. 5a). Previous work demonstrated the regulatory role of ribosome pausing in the processing of ubiquitin peptides[40]. Here we identified five periodic disome peaks in the ubiquitin coding gene *UBI4l*, with a novel peak at the beginning of the gene, comparing to the training dataset (Fig. 5b). All the peaks were positioned at a proline-rich motif (PPD). When all the disome and monosome profiles are aligned based on the PPD motif, the disome profiles show clear periodic peaks upstream of the pause sites, which is not apparent in the input monosome profiles (Fig. 5c).

## Riboformer allows interpretation of exacerbated ribosome stalling in aging

High levels of ribosome collisions can lead to proteostasis collapse in aged organisms[13]. To investigate the mechanism of aging-related ribosome pausing, we applied the Riboformer pipeline to ribosome profiling data from young and old yeast cells[13]. Using ribosome profiles in young yeast (day 1) as the control, our pipeline successfully predicted ribosome occupancy in aged yeast (day 4, $r = 0.94$, Supplementary Fig. 9a, b). In silico mutagenesis analysis of the aging-related pausing sites ($n = 6347$, Supplementary Fig. 9c) identified a few clusters with a low SIS at the ribosome decoding site. Further examination of these clusters revealed significant enrichment of Pro codons in the ribosomal A site (Supplementary Fig. 9e, f). This observation was not discernible upon analysis of all the ribosome pausing sites[5]. We further extended our analysis to the aging experiments in worms

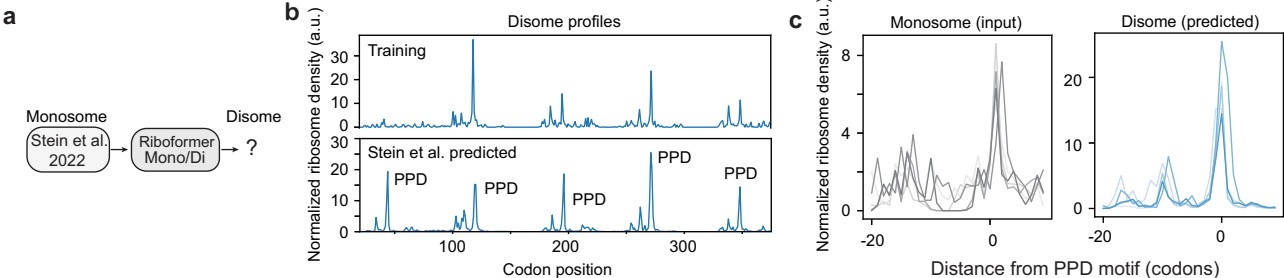

**Fig. 5 | Riboformer predicts disome sites in unseen dataset. a** The trained Riboformer model is used to predict the disome profiles in the Stein et al. dataset[5]. **b** The monosome and predicted disome profile for the gene *UBI4* in the training dataset (top) and the Stein et al. dataset (bottom). **c** The monosome profiles before the five PPD ribosome pausing sites (left), and the predicted disome profiles before the pausing sites (right) are shown for the gene *UBI4*. Source data are provided as a Source Data file.

(*Caenorhabditis elegans*, Supplementary Fig. 10). Interestingly, when we examined SIS for the age-dependent pause sites ($n = 8376$, Supplementary Fig. 10c), there was an enrichment of Asp codon in the P site for the clusters with similar shapes (Supplementary Fig. 10d–f). This observation agrees with the enriched motifs associated with age comparisons (day 12 vs. day 1) in the original paper[13]. In both aged yeast and worm, the SIS was positively correlated with the number of positively charged amino acids (Supplementary Figs. 9h and 10h), unlike what we observed with yeast disomes. Interestingly, the overloaded RQC pathway in aging organisms does not target highly positively charged protein sequences[48], which may explain the observed positive correlation.

In our analyses of yeast disomes described above, we observed a negative correlation between mRNA folding energy and SIS, indicating that the ribosomes are more likely to pause in structured regions of mRNA. This correlation holds true for predicted ribosome density from day 4 yeast cells (Supplementary Fig. 9g, $r = -0.84$). Surprisingly, SIS and mRNA folding energy were *positively* correlated in the ribosome collision sites in aged worms (Supplementary Fig. 10g, $r = 0.97$). Our results imply that mRNA secondary structures might play different roles in aging-related ribosome stalling events in these model organisms. Overall, our approach provides a general pipeline for the interpretation of context-dependent ribosome pausing and reveals novel insights into how local context affects aging-dependent translation dynamics.

## Discussion

Taken together, our work presents a general predictive framework for standardizing and interpreting ribosome profiling experiments across different organisms and experimental conditions. Our framework models the change in ribosome kinetics caused by the experimental protocol, offering a unique opportunity to correct protocol biases in pre-existing datasets and circumvent the need for certain resource-intensive experiments in standard protocols. We have benchmarked its performance by removing experimental artifacts resulting from rapid filtering and the Cm-containing lysis buffer across 16 ribosome profiling datasets produced by four different labs. We anticipate that our method will also be useful in clarifying ribosome density in eukaryotic samples. While most yeast protocols use cycloheximide to arrest translation in the lysis buffer, Wu et al. found that adding cycloheximide and tigecycline together yields short footprints (~21 nt) from ribosomes with empty A sites and longer footprints (~28 nt) with full A sites[49]. We further demonstrated that Riboformer can be trained to predict these short and long-footprint distributions from libraries created with the cycloheximide-only protocol, indicating Riboformer's broad applicability to existing ribosome profiling datasets (Supplementary Table 5). Finally, Riboformer can be trained on any pair of ribosome profiling datasets. This flexibility enables in silico extrapolation of ribosome densities using a limited number of existing data.

Using a trained model to estimate disome profiles based on monosome data, our method can even predict new disome peaks that do not exist in the training datasets.

By simulating the impact of sequence mutations on ribosome occupancy, the Riboformer model identifies the sequences responsible for ribosome collisions, providing insights beyond simple motif analysis. This approach enables a granular classification of ribosome pausing sites, uncovers the impact of amino acid charges and mRNA structure on ribosome collisions and identifies the effect of proline-enriched motifs on ribosome stalling in young and aged yeast. Moreover, it provides insight into the regulatory code of translation kinetics, facilitating the discovery of novel therapeutic targets. For example, we applied Riboformer to analyze the ribosome profiles of SARS-CoV-2 following infection of human cells[50]. Our findings reveal that binding motifs of fragile X mental retardation protein (FMRP) contribute to the increased ribosome occupancy in later stages of infection (Supplementary Fig. 11). Notably, FMRP has been demonstrated to bind to polysomes[51], and our observation implies the therapeutic potential of Fragile X syndrome drugs for inhibiting SARS-CoV-2 viral reproduction. Interestingly, the antiviral activity of FMRP has been reported for ZIKA virus[52]. In addition, a new study reveals that the SARS-CoV-2 virus load is reduced with the inhibition of mGluR5, a leading drug target for Fragile X syndrome that signals through FMRP[53].

The Riboformer framework is not without its limitations. Firstly, it relies on existing datasets for training. With the development of techniques for unbiased measurement of translational landscape, we envision that new Riboformer models can be further trained to improve the analysis of biased datasets. In addition, like many existing methods, Riboformer does not consider translation initiation and termination, both of which can affect ribosome queuing along the transcript. Our model excludes the first and last ten codons in the gene coding region in the downstream analysis. This could be addressed in future work through systematic quantification and modeling of translation initiation and elongation rates. Thirdly, Riboformer is not designed to handle rare events like ribosomal frameshifting, due to the limited number of training samples. To tackle these specific situations, transfer learning approaches could be explored, which allows for initial training on one task and subsequent fine-tuning across various contexts. Finally, while our SIS analysis identifies specific ribosome stalling sites that could be mediated by sequence features such as proline-rich motifs, further experimental work will be needed to expand on these findings.

Nonetheless, our Riboformer model distinguishes experimental artifacts from real biological signals and provides a means for the integrated analysis of existing heterogeneous ribosome profiling datasets. Comparison of ribosome profiles across multiple species allows the study of ribosome stalling through the lens of evolution, paving the way to investigate the evolutionary forces that determine codon selection and elongation efficiency. Further, with the rapid development of single-cell sequencing methods such as single-cell

Ribo-seq and RIBOmap[54,55], context-aware models like Riboformer will make it possible to study translation dynamics in a cell state and cell type-specific manner. Riboformer can be used as a pure sequence-based model when the reference input is masked, or in combination with other computational methods such as *Scikit-ribo*[56] and *choros*[57] to enable a more accurate estimation of ribosome distribution. While primarily developed for the ribosome profiling datasets, we envision the Riboformer pipeline could be widely applicable for modeling the experimental bias and biological variations in other types of high-throughput sequencing data.

## Methods

### Ribosome profiling datasets

The ribosome profiling dataset for *E. coli* cells (Cm vs. high-Mg lysis buffer) was obtained from the NCBI GEO database with accession number GSE119104. The Burkhardt et al. dataset was obtained from the NCBI GEO database with accession number GSE77617. The ribosome profiling dataset for genetic circuits was obtained from the NCBI GEO database (GSE152664). Genomic data, including gene sequences, as well as transcript and open reading frame (ORF) boundaries, were obtained from NCBI. The *S. cerevisiae* and *C. elegans* aging datasets were downloaded from NCBI GEO (GSE152850). Monosome and disome profiles were obtained from NCBI GEO (GSE139036). The ribosome profiles of SARS-CoV-2 were obtained from NCBI GEO (GSE149973). For all the ribosome footprint experiments, we excluded the first and last 10 codons in the downstream analysis to avoid the atypical footprint counts observed at the beginnings and ends of genes. To model ribosome density without being biased by the heterogeneity of translational speed along the 5′ ramp and to obtain robust estimates of the steady-state distribution, we excluded all the genes with length <200 nt. In addition, we filtered out genes with poor ribosome coverage, in accordance with previous works[5,12]. Genes with fewer than 0.5 reads per nucleotide on average in prokaryotes and genes with fewer than 5 reads per nucleotide on average in eukaryotes were excluded from the analysis. For ribosome profiling experiments with replicates, the mean ribosome occupancy at each nucleotide is used for the following analysis. For the codon of interest, we calculated the pause score by taking the mean ribosome density in the 3nt window and dividing it by the mean density across the ORF. The pause scores of codons represent the mean of the scores for all instances of the codon of interest. We further *z*-score normalized the codon pause scores before visualization.

### Implementation and architecture of Riboformer

We used the RNA sequence and the normalized ribosome density in the control experiment as a separate input to the Riboformer model. For both inputs, our model employs 5 convolutional blocks and 1 transformer block (see below for more details) to extract the features of coding sequences and reference ribosome densities. We used an element-wise multiplication layer to pool the information from the two branches together followed by a feedforward layer that produces the model output.

The 40-codon sequence is the input for the first branch, and it was further transformed into a vector using sequence embedding (hidden dimension: 8). For the input sequence $x \in R^{L \times E}$ (length $L$ across $E$ dimensions), the first stage of the architecture aims to extract the relevant sequence motifs from the mRNA sequence using the following block of operations:

1. 2D convolution, with the kernel size of (5, 5), filter number 32:

$$x_i^f = \sum_m x_{i+m} \cdot K_m^f + b_m^f, \tag{1}$$

where $K_m^f$ and $b_m^f$ are the learnable weight and bias matrices of the *f*th filter.

2. Batch normalization.
3. Rectified linear activation unit (ReLU) activation:

$$x_i^f = \text{ReLU}(w^{fc} \cdot x_i^f), \tag{2}$$

where $w^{fc}$ stands for the learnable weights for the fully connected layer.

We applied this block 5 times. Then the information from all 32 filters was pooled together using average pooling.

The second stage of the architecture aims to capture the interdependency among the codons, similar to many natural language processing tasks. We used a multi-head attention (MHA) layer. Given an input sequence $x \in R^{L \times C}$ (length $L$ across $C$ channels), each attention head has a set of weights $w^q \in R^{C \times K}, w^k \in R^{C \times K}$, and $w^v \in R^{C \times K}$. These weights transform the input sequence into queries, keys, and values, defined as

$q_i = x_i \cdot w^q, k_i = x_i \cdot w^k$, and $v_i = x_i \cdot w^v$.

The attention matrix is then derived from the equation:

$$a_{ij} = \text{softmax}(q_i k_j^T / \sqrt{K}), \tag{3}$$

Here $a_{ij}$ represents the influence of the query at position i on the key at position j. The values depict the data each position contributes to the subsequent positions attending to it. Each single attention head computes its output as a weighted sum over all input positions: $h_i = a_{ij} \cdot v_j$. This mechanism enables each query position to access information from the entire sequence. The multiple heads compute with independent parameters, and their outputs are concatenated to yield the final layer output. Our layers used 10 heads, key/query dimension of 8, and a dropout rate of 0.1. The last feed-forward module uses a fully connected layer followed by layer normalization and the ReLU activation function.

The input to the second branch is the ribosome density of the same codon sequence from the control experiment (40 codons). For each codon, we calculated the sum of reads from all three nt. Then the ribosome density is further log-transformed and processed by a neural network structure that is similar to the first branch. The only difference is that 1D convolution layers are used in the convolution block, instead of 2D convolution layers. The output of the second branch has the same dimension as the first branch (32). Finally, element-wise multiplication was used to combine all the information from the two branches, and a ReLU activation function was used to predict the ribosome density at the position of interest in the new condition: $x_{\text{output}} = \text{ReLU}(w^{fc} \cdot x_{\text{coding}} \cdot x_{\text{ref}} + b)$. The model was implemented in Tensorflow and the source code is available at https://github.com/lingxusb/Riboformer.

### Riboformer training and hyperparameter tuning (training and validation dataset construction)

Adam optimizer was used to train the Riboformer model on an A100 GPU (40 GB, Nvidia). A cosine learning decay was used to schedule the learning rate with a start learning rate of 0.0005:

$$\text{learning rate} = 0.0005 * \frac{1 + \cos(\pi * \text{step})}{2}, \tag{4}$$

The mean squared error loss function was employed to measure model performance in both the training and validation stages. The explanatory input data and corresponding response variables were divided into training (70%), validation (15%), and test (15%) sets. Early stopping was introduced to prevent overfitting, and the training process terminated when the validation loss did not decrease for 10 epochs. For building and training models, Keras v2.2 and Tensorflow v1.10 software packages were used.

## Codon positional enrichment

We calculated the translation efficiency (TE) for a target gene as the ratio between the mean of the ribosome density (RD) and the mRNA expression. The ribosome density (RD) of each gene was calculated by averaging all ribosome occupancies over the length of the gene[20]. The mRNA expression in FPKM (fragments per kilobase of transcript per million mapped reads) of each gene was calculated by averaging the height of the RNA-seq profile over the length of the gene.

We analyzed the first 100 codons of genes with TE in the highest/lowest 10 percentiles among all the genes. In each 10-codon window, we calculated the number of a specific codon. It is then compared with the codon number from a randomly sampled gene group with the same number of genes. We calculated the p values from a student t-test (function *ttest_ind* from the scipy package) as the positional enrichment for the specific codon.

## Identification of conditional-dependent pause sites

To identify conditional-dependent ribosome pausing sites, we used a strategy that is similar to Stein et al.[5], which utilized two-tailed Fisher's exact tests to identify codon positions with statistically significant changes in ribosome pausing. At each codon position, $2 \times 2$ contingency tables were created to perform a two-tailed Fisher's exact test to compare the ratio of the reads in the control sample and the sample of interest. This compares the observed ratio of ribosome reads at a specific position from the two samples to the expected ratio based on the total number of reads from the two samples. It allows the calculation of the odds ratio as well as the *p*-value. The first 10 and last 10 codons of the transcript were excluded in the analysis. The conditional pausing sites were identified as follows: *p*-value < 0.001 and odds ratio > 1.

## In silico mutagenesis analysis

For each conditional dependent pausing site, we denote the Riboformer predicted ribosome density as RD. In the 40-codon input sequence, we selected a 10-codon window and sampled 100 random sequences $\{x_j\}$ to replace the original sequence. The mean predicted ribosome density from the random sequences was calculated as $RD' = \frac{1}{100} \sum_{j=1}^{100} RD(x_j)$ and $RD - RD'$ is the sequence impact score (SIS) for the 10-codon window. We moved the window along the RNA sequence at one codon step so that every codon was randomly mutated 1000 times. Enrichment of known sequence motifs of RNA-binding proteins was identified using SEA[58].

## Clustering analysis of sequence impact scores

We used the *K*-means clustering method from the Python scikit-learn package to cluster the impact score profiles. Elbow method was used to determine the cluster number and the random seed was set to 0.

For each 40-codon sequence, we calculated its folding energy using the RNAfold software (https://www.tbi.univie.ac.at/) with default parameters. The energy was then averaged for each cluster.

To calculate the codon enrichment for each cluster, the codon occurrences at each position (−20 to 20) for each cluster were compared with randomly sampled codon sequences. A Student *t*-test was used to calculate the *p*-value of the enrichment or depletion of the specific codons. The sequence log was generated based on the log-transformed *p* values.

## Reporting summary

Further information on research design is available in the Nature Portfolio Reporting Summary linked to this article.

## Data availability

We provide all datasets generated or analyzed during this study. The ribosome profiles were downloaded from Gene Expression Omnibus with the accession numbers GSE119104 (Mohammad dataset[12]), GSE77617 (Burkhardt dataset[16]), GSE98664 (synthetic circuit dataset[20]), GSE152850 (aging dataset[21]), GSE139036 (disome dataset[5]), GSE149973 (SARS-CoV-2 dataset[50]), and GSE115162 (Wu dataset[49]). More information for these datasets can be found in the "Methods" section. Source data are provided with this paper.

## Code availability

Codes for the Riboformer pipeline are available from https://github.com/lingxusb/Riboformer and https://doi.org/10.5281/zenodo.10594484. Codes for reproducing the figures including Figs. 2d, e, 4b, c, e, Supplementary Figs. 4 and 5, are available from GitHub (https://github.com/lingxusb/Riboformer/tree/main/reproducibility).

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

## Acknowledgements

We are grateful to Y. Yan, R. Majovski, H. Kang, J. Sternberg, A. Vieira, and N. Guydosh for their helpful discussions and all the reviewers for their constructive feedback. B.S. acknowledges the help of Broad Communications Lab. This work was supported by the NIH grant GM136960 to A.R.B., and the National Key Research and Development Program of China (2021YFF1200500) to L.L.

## Author contributions

B.S. conceived the research project and designed the neural network model. B.S. and J.Y. implemented the model and carried out model training and validation tasks. B.S. performed the computational and statistical analyses. L.L. and Y.C. provided computational expertise and input. B.S., J.Y., J.Z., and A.R.B. wrote the manuscript. All the authors discussed the results and commented on the manuscript.

## Competing interests

The authors declare no competing interests.
