## [Peer Review File · Nature Communications]

Riboformer: A Deep Learning Framework for Predicting Context-Dependent Translation DynamicsREVIEWER COMMENTS

Reviewer #1 (Remarks to the Author):

This study introduced an improved method when evaluating the density of ribosome footprints from ribosome profiling data. Ribosome profiling relies on deep sequencing of the RNA fragments that were protected by ribosomes during RNase digestion to study the positions of ribosomes during translation. From the sequencing results, the density of ribosome footprints on codons is often used to estimate the relative speed of elongation in a local context. However, the interpretation of the analysis results depends on the accurate measurement of footprints at codon resolution. Indeed, there has been a lack of standardized data-processing methods preventing uniform interpretation of analysis results among different datasets. To overcome these limits during data analysis, the authors proposed a deep-learning-based framework to estimate the density of ribosome footprints, harnessing the transformer algorithm. This method was expected to minimize the biases originating from library preparation and highlighted the translational pauses that happened during stresses. Before the publication, the authors should address the following concerns.

Major points

1. This reviewer wondered when and how the Riboformer is most useful in regular ribosome profiling experiments. In bacteria, as clearly indicated by Mohammad et al. eLife 2019, the high Mg condition has been the best condition to monitor translation. So researchers should simply consult this work and followed the protocol that the paper suggested. In this case, Riboformer may not be necessary to correct the data. A similar logic can be applied to disome profiling data; although the experiment per se may be time and cost-consuming, researchers could do disome profiling to survey ribosome pause sites, without the conversion from monosome profiling via Riboformer. In terms of this, the effective situation of Riboformer could be limited in handling the pre-existing data, but not the experiments that will be designed in the future.

2. Related to the point above, this reviewer also wondered how SIS analysis is useful. This reviewer guessed that thanks to the training, authors could test the impact of sequences in silico. However, the same calculation could be done by the simple motif analysis of the training data, without Riboformer conversion. Again, simply, better conditions of ribosome profiling (high Mg in bacteria and disome profiling) will be more direct than training and SIS analysis by Riboformer. The authors should highlight the effectiveness and usefulness of the SIS analysis practically.

3. This reviewer wondered how many replicates are necessary to train the Riboformer. For example, in Figure 1c, the authors used a single dataset of flash-freezing with a lysis buffer of high Mg for training. The author should show the robustness and reproducibility of this training, using a couple of replicated data for training and test whether similar results are obtained.

4. The examples shown in Figure 1 were related to the technical issues in bacterial ribosome profiling. On the other hand, the application shown in Figure 2 was more related to the biological interpretation of the data. These two themes were totally different but not well introduced in the manuscript. The authors should consider more helpful and careful explanations of these subjects.

5. In Supplementary Figure 5b and Supplementary Figure 6c, at the disome pause site, there was a complete absence of monosome footprints, which was not normally expected. Could the authors provide more explanation or discussion about these observations?

Minor points

1. In Supplementary Figure 7f, this EPA codon enrichment result appears to be in apparent contradiction with the results of the original paper (Stein et al. Nature 2022). Explanation regarding this will be helpful for readers.

2. In Supplementary Figure 8, the authors should discuss whether FMRP binding to mRNA and the subsequent translation modulation is beneficial or detrimental to SARS-CoV-2 infection.

3. At line 59, "The" should be "the".

4. Some of figure citations should be corrected:

At line 110, "Fig 1f" should be Fig. 1e;

At line 118, "Fig 1g" should be Supplementary Fig 2;

At line 114, "Fig. 1h" could not be guessed which figure should be cited here (but definitely not Fig. 1h);

At line 126, "Fig 1g" should be Fig. 1f and 1g.

5. At lines 146-147, the authors described Trp codon enrichment in the first 80 codons in low TE genes (Supplementary Figure 4). However, the meaning of these data was not clear. Authors should consider removing these data, or alternatively provide a discussion/explanation of how these data should be interpreted.

Reviewer #2 (Remarks to the Author):

In this manuscript, Shao et al developed Riboformer, a deep learning method to model ribosome protected reads density, and showed a few of its applications. Though deep learning modeling ribosome profiling reads is an interesting topic, the manuscript does not appear to be well written in general. Its overall logic is not clear to readers. It lacks sufficient details and falls short in explaining the results of in silico modeling. Moreover, additional experiment validations are required to support the in silico findings. Overall, the manuscript might not be suitable for publication without plenty of additional work.

Major point:

1. The performance of Riboformer would need to be evaluated with independent test data (i.e. AUROC, AUPRC, and the correlation between the modeling density and the true density)

2. A comprehensive comparison of different existing methods of modeling ribosome protected reads such as ROSE, RiboMIMO, Riboexp (PMID: 28957655, 33770074, 33479731) would be needed.

3. In Fig2, authors identified sequence determinants of ribosome pausing as a new finding. The related experimental validations should be provided to support these modeling findings. The same causality-validation mindset might be applicable to a few other in silico modeling findings in the manuscript.

4. The manuscript should be re-written clearly to show its full logic and details to readers.

Reviewer #3 (Remarks to the Author):

In this manuscript Shao et al describe software called Riboformer for the analysis of ribosome profiling data. The software is based on the use of convolutional neural networks (CNN). In essence Riboformer can compare ribosome profiling densities for a set of genes from two datasets (say A and B), learn what sequence features (within 120 nt window) are responsible for the difference between A and B

and then use this knowledge to predict how ribosome densities would look in B given data from A. This is a highly useful functionality as demonstrated by the authors. Previously Buskirk and colleagues (ref 16) have developed an experimental approach for obtaining ribosome profiling to minimise the distortions in ribosome density introduced by experimental procedures (e.g. drug chloramphenicol treatment, culture filtering and scrapping). This procedure is laborious and is hard to implement. By using Riboformer one could use an easier protocol that produce less accurate data and use Riboformer to predict how the data would look like if they were obtained with the "unbiased" protocol. The authors also successfully demonstrated the utility of Proetoformer as data transformation tool to predict disome densities from monosome profiling and to predict pauses expected in aged cells. Furthermore, the authors came up with a clever approach to look into the "black box" of Riboformer's CNN. For this they performed an in silico mutagenesis and explored mutations that influence pausing, thus figuring out what sequence features are expected to cause pauses. So overall, Riboformer is a very useful addition into a set of computational tools for ribosome profiling data analysis, I can clearly see how it can be used by others and for additional applications. However, I have several critical comments and suggestions for improvements.

General criticism

1. First of all, we were not able to install and run software, even though tried several different computers through conda on both in Ubuntu 22LTS and Windows 10. Some error logs are at the end of this report. The authors should test Riboformer on several different platforms and try to maximise its compatibility. Perhaps it makes sense to explicitly describe environment requirements. To make the tool useful it should be reasonably easy to install and run.

2. Although the manuscript is written in good language it describes Riboformer's capabilities in terms that are too generic. The limitations of Riboformer also not clearly stated, as a result it was not easy to understand what it specifically does. I suggest trying to use more specific language. It will make it easier for user and readers to understand what Riboformer does and what should it be used for. Several specific points below are illustrations.

Specific critical points and suggestions:

3. The abstract states that Riboformer "corrects experimental artefacts in previously unseen datasets". Also on lines 133-134 "Riboformer is a predictive framework that can be used to standardize a wide range of ribosome profiling measurements, reducing experimental noise while remaining true to the underlying biological signal of interest."

Strictly speaking Riboformer alone does not do this. If we have an unbiased dataset Proetoformer can learn how to remove the biases in the biased datasets. So the performance of Riboformer is critically dependent on the existence of the improved experimental datasets. This point is not communicated clearly in the manuscript. Even figure 1 gives an impression that Riboformer takes as input ribosome profiling dataset and a sequence and somehow magically produces unbiased dataset. But this is a false impression because Riboformer can do it only if an "unbiased" dataset is given for training (hence it should be shown as a part of input). In the specific examples used by the authors, Riboformer is used to convert data obtained with Chloramphenicol to what would be expected if the data were obtained with blocking the ribosomes with high Mg concentration. If high Mg concentration was perturbing ribosome locations, Riboformer would introduce (rather than remove) the experimental biases. It would be helpful if the authors could also demonstrate RiboFormer performance to remove biases arising from drug treatment artefacts using the biased (chx) and unbiased (chx/tig) data from Wu et al 2019 doi: 10.1016/j.molcel.2018.12.009.

4. Another abstract statement Riboformer "uncovers a bottleneck in protein synthesis", also the title of one of the subsections "Riboformer allows identification of limiting steps in protein synthesis". This statement is too generic and obscure, many people would think of translation initiation and availability

of certain initiation factors as bottlenecks in protein synthesis. Here, however, the authors reported finding Trp deficiency that effects progression of the ribosomes during the elongation state. To what extent this deficiency effects the rate of protein synthesis is not yet clear. A more specific description of the finding would avoid potential misinterpretation of the claim regarding the bottlenecks.

5. Some descriptions are vague and as a result appear contradictory, i.e. "that detects long-range dependencies in the regulation of elongation (Fig. 1a)." and "More specifically, our approach assumes that the relative change in ribosome occupancy is primarily determined by the surrounding sequences." So is it long distance or surrounding? In practice, the authors explore a window of 120 nucleotides centred at the A-site of the footprint generating ribosome. I think the use of "long-range dependency" here may be misleading as some may think of much longer intervals. The choice of 120 nucleotides sounds reasonable intuitively, it takes into account the sequence encoding the polypeptide inside the ribosome that may influence its progression as well as downstream sequence responsible for interfering RNA structures and effects of downstream ribosomes. However, this appears as an arbitrary choice. Did the authors play with the length of the window? It would be nice to explore how the size of the window affect the accuracy of transformations.

6. The data used for training and testing were processed to include only highly covered transcripts. Although this approach improves signal-to-noise ratio, it also may potentially introduce biases specific to highly expressed transcripts. I wish the authors explored this a little more. How well RiboFormer works on predicting densities at the lowly expressed transcripts? How specific coverage thresholds influence its accuracy? The current threshold seems a bit odd, for prokaryotes it is 0.5 reads per nt and for eukaryotes 5 reads per nt. I would have thought it should be the way around since prokaryotic riboseq data usually exhibit higher coverage due to the small size of their genomes.

7. I think the authors should explicitly describe the limitations of the software. In my opinion these are (i) Reliance on unbiased or condition-specific datasets. (ii) Limited-range dependencies. It is reasonable to assume that certain ribosome behaviours such as ribosome queuing would depend not only the strength of ribosome stalling, but also on the rate with which ribosome initiate on mRNA. That, however, is excluded from the modelling. (iii) The modelling is limited to middle parts of coding regions, excluding regions upstream and downstream and therefore it is unclear whether the transformation would give reasonable results in those regions. (iv) Like with some other previous approaches such as RUST, the model likely captures general factors effecting ribosome dwell times on all or most mRNAs and as a result it may fail to work adequately on special and unique situations, such as ribosomal frameshifting, exceptional stalling sites like, etc. This could be potentially explored, but may not be easy, as one first needs to identify a pair of datasets where experimental or biological conditions effect these rare events.

8. It would be good if the authors provide more explicit detail on what data need to be provided for Riboformer. Wig files are required, but what should be used as coordinates? Presumably offsetted locations of A-sites, but this is not stated. How the data should be processed? It is likely that offsetting may affect the outcomes (i.e. read length dependent offsetting vs uniform offsetting), for a novice users some instructions could be very useful in this regard and will help avoid frustration.

9. Pearson coefficients are used as a measure of correlations, but those are very sensitive to outliers and unclusion/omission of a few datapoints may significantly alter the correlations. For the prudency it may be helpful to include Spearman correlations.

10. Code for most subfigures is not available, but some are stated in reproducibility section of pipeline, mention this explicitly in Code availability section.

11. Sup. Figure 6c text: spelling error: . n = 6,347 sites.

Error log:

- Installation failed through conda (ubuntu 22), I had to remove line:

"- json=0.0.1=0" in yml file for it to work.

- Program fails to run using example data:

```
cd Riboformer/Riboformer
```

```
python3 transfer.py -i ../datasets/GSE139036_disome -m ../models/yeast_disome
```

Error:

Data Loading: 0file [00:00, ?file/s]

Traceback (most recent call last):

File "/home/roler/Desktop/forks/Riboformer/Riboformer/transfer.py", line 86, in <module>
main()

File "/home/roler/Desktop/forks/Riboformer/Riboformer/transfer.py", line 65, in main

```
x_c[:, :40] = x_c[:, 0:40] / 100 - 5
```

TypeError: 'NoneType' object is not subscriptable

Pasha Baranov

Reviewer #1 (Remarks to the Author):

This study introduced an improved method when evaluating the density of ribosome footprints from ribosome profiling data. Ribosome profiling relies on deep sequencing of the RNA fragments that were protected by ribosomes during RNase digestion to study the positions of ribosomes during translation. From the sequencing results, the density of ribosome footprints on codons is often used to estimate the relative speed of elongation in a local context. However, the interpretation of the analysis results depends on the accurate measurement of footprints at codon resolution. Indeed, there has been a lack of standardized data-processing methods preventing uniform interpretation of analysis results among different datasets. To overcome these limits during data analysis, the authors proposed a deep-learning-based framework to estimate the density of ribosome footprints, harnessing the transformer algorithm. This method was expected to minimize the biases originating from library preparation and highlighted the translational pauses that happened during stresses. Before the publication, the authors should address the following concerns.

We thank the reviewer for a nice summary of our work.

Major points

1. This reviewer wondered when and how the Riboformer is most useful in regular ribosome profiling experiments. In bacteria, as clearly indicated by Mohammad et al. eLife 2019, the high Mg condition has been the best condition to monitor translation. So researchers should simply consult this work and followed the protocol that the paper suggested. In this case, Riboformer may not be necessary to correct the data. A similar logic can be applied to disome profiling data; although the experiment per se may be time and cost-consuming, researchers could do disome profiling to survey ribosome pause sites, without the conversion from monosome profiling via Riboformer. In terms of this, the effective situation of Riboformer could be limited in handling the pre-existing data, but not the experiments that will be designed in the future.

We thank the reviewer for these thoughtful comments. We would like to clarify that Riboformer offers several unique functionalities: firstly, it simplifies experimental protocols and lowers the technical barrier for more complicated experiments; secondly, Riboformer predicts ribosome distribution in novel contexts using a limited number of experiments; lastly, we believe the analysis of pre-existing datasets remains essential for generating biological hypotheses, and we have demonstrated that Riboformer is widely applicable across both prokaryotic and eukaryotic datasets. However, in light of the reviewer's comments, we acknowledge that relying on pre-existing data for model training is one limitation of our method, and we added this discussion in the revised manuscript.

(1) Riboformer simplifies experimental protocols.

As reviewer #3 has pointed out, even though we have developed an experimental approach that minimizes the distortions in the measured ribosome density, this protocol can be laborious and challenging to implement. In this case, Riboformer can be trained on unbiased dataset and allows researchers to approximate the ribosome profiles from more

complicated protocols using simpler ones, thereby circumventing the need for certain resource-intensive experiments.

(2) Riboformer predicts ribosome distributions in new contexts.

As the reviewer pointed out, experiments can be both time and cost consuming. Riboformer is a user-friendly package that can be trained on any pair of datasets. This flexibility enables *in silico* extrapolation of ribosome densities using a limited number of existing data. For example, **Fig. 5** illustrates that Riboformer can estimate disome profiles from monosome data, and even predict new peaks that are not existing in the training datasets.

(3) Riboformer improves analysis of pre-existing ribosome profiling data.

The vast number of existing ribosome profiling datasets presents a rich resource for harnessing biological insights and forming new hypotheses. However, most of them are plagued by various technical artifacts. Through case studies involving four datasets generated by different labs (**Fig. 2, Fig.3, Supplementary Fig. 4 and Fig. 5**), we showed that Riboformer effectively removes technical bias while preserving the true biological signal. Beyond bacteria cells, we also conducted new computational experiments to show that Riboformer removes technical bias arising from antibiotic treatment in yeast (**Supplementary Table 5**).

To the best of our knowledge, none of the above functions can be achieved with existing deep learning-based methods. Additionally, we have demonstrated the usefulness of SIS analysis, which can provide novel insights into the regulation of translation elongation (**response to Q2**). Our modifications are reproduced in the box below.

Revised text (Section “Discussion”):

The Riboformer framework is not without its limitations. Firstly, it relies on existing datasets for training. With the development of techniques for unbiased measurement of translational landscape, we envision that new Riboformer models can be further trained to improve the analysis of biased datasets.

2. Related to the point above, this reviewer also wondered how SIS analysis is useful. This reviewer guessed that thanks to the training, authors could test the impact of sequences *in silico*. However, the same calculation could be done by the simple motif analysis of the training data, without Riboformer conversion. Again, simply, better conditions of ribosome profiling (high Mg in bacteria and disome profiling) will be more direct than training and SIS analysis by Riboformer. The authors should highlight the effectiveness and usefulness of the SIS analysis practically.

We would like to thank the reviewer for this feedback concerning the practical application of SIS in ribosome profiling experiments. SIS analysis provides capacities that are beyond simple motif analysis or better conditions of ribosome profiling experiments. Briefly, it allows a granular classification of ribosome pausing sites, identifies potential mechanisms of

ribosome pausing and provides insights on the role of RNA binding protein in viral replication.

(1). Riboformer allows a granular classification of ribosome pausing sites.

While motif analysis provides an overview of the ribosome pausing sites, Riboformer allows the in-depth classification of these sites based on the sequence determinant. For example, simple motif analyses of the disome profiling experiment reveal the enrichment of Proline at the disome formation sites (**Fig. R1b**). However, a more careful analysis with Riboformer reveals clusters with diverse codon enrichment profiles. We found cluster 9 and 10 are enriched in triple proline motifs, while Trp and Lys codons are enriched in cluster 4 and 5 (**Fig. R1a**).

(2). Riboformer identifies potential mechanisms of ribosome pausing.

Identification of sequence determinant for specific ribosome stalling events is hard since it requires mutagenesis of specific sequences, followed by measurement of changes in ribosome densities. Riboformer provides an efficient tool for pinpointing potential mechanisms of ribosome stalling. For example, we found a low SIS score for cluster 4 after the ribosome decoding site. Interestingly, cluster 4 also has the highest structure score in this region (**Fig. R1c**), suggesting that ribosome stalling for these sites could be mediated by mRNA folding in the downstream sequence. In addition, consecutive Lys codons could be potential sites for ribosome collision (Stein et al. Nature 2022) and our algorithm identified it as the sequence determinant of disome peaks in the PWP1 gene (**Fig. R2**).

(3). Riboformer reveals the potential role of RNA-binding protein in viral replication.

Analysis of COVID-19 datasets with Riboformer revealed an enrichment of FMRP motif in sequences with low SIS scores, suggesting its potential implication in viral replication. FMRP is a mRNA binding protein that associates with polysomes, and our finding is consistent with prior research on viruses like ZIKA (Soto-Acosta et al. eLife 2018), in which the depletion of FMRP elevated viral titers and rate of infection (see response to minor point 1).

In summary, our SIS analysis identifies specific ribosome stalling sites that could be mediated by mRNA secondary structure, RBP binding and polybasic stretches such as consecutive Lys codons. It also reveals distinct codon enrichment profiles than the simple motif analysis of all the pausing sites. We have incorporated these clarifications in our manuscript, which is also reproduced in the box below.

Revised text (“Riboformer identifies sequence determinants of ribosome collisions”):

Cluster 4 and 5 show distinct SIS profiles from the population average, with Trp and Lys codons enriched at the ribosomal A site, respectively (Supplementary Fig. 7). We found pausing sites from cluster 4 are more likely to be affected by the mRNA secondary structure of the downstream sequences (Supplementary Fig. 7c). In addition, previous works have demonstrated that consecutive Lys codons (polybasic region) could be potential sites for ribosome collision^{7,36}. Riboformer further identifies consecutive Lys codons as the sequence determinant of disome peaks in the PWP1 gene (Supplementary Fig. 8).

In summary, our interpretable framework identifies the sequences responsible for ribosome collision events, clusters these sequences into distinct groups, and uncovers various modes of ribosome stalling, offering insights beyond motif analysis of all the pausing sites.

References:

Soto-Acosta R, Xie X, Shan C, Baker C, Shi P, Rossi S, Garcia-Blanco M, Bradrick S (2018) Fragile X mental retardation protein is a Zika virus restriction factor that is antagonized by subgenomic flaviviral RNA eLife 7:e39023.

Figure R1. SIS analysis of disome formation sites. **a**, SIS profiles of all the disome peaks are grouped into 10 clusters (top), and codon enrichment profiles for cluster 4 (n = 675 sites), cluster 5 (n = 497 sites), and cluster 10 (n = 272 sites) are shown (bottom). **b**, codon enrichment profile for all disome peaks (n = 11,079 sites). **c**, mean folding energy of the input mRNA sequence for all clusters over a 30 nt window (+15 nt to +45 nt). This figure is labeled as **Supplementary Figure 7** in the revised manuscript.

Figure R2. SIS analysis identifies consecutive Lys codons as the sequence determinant of disome peaks. Monosome (gray, top) and disome (blue, middle) profiles of the PWP1 gene in WT yeast cells are shown. The SIS profile (bottom) around the disome peaks is shown and the region with consecutive Lys codons is shaded in light gray. This figure is labeled as **Supplementary Figure 8** in the revised manuscript.

3. This reviewer wondered how many replicates are necessary to train the Riboformer. For example, in Figure 1c, the authors used a single dataset of flash-freezing with a lysis buffer of high Mg for training. The author should show the robustness and reproducibility of this training, using a couple of replicated data for training and test whether similar results are obtained.

We thank the reviewer for raising this point. To fully address this concern, we used the two replicates from yeast and *C. elegans* aging dataset (Stein et al. Nature 2022) to train and test our models. We trained the Riboformer models using each replicate individually and the mean ribosome density from both. To evaluate model performance, we conducted 3-fold cross validation tests: all the input genes were randomly split in 3 groups. In each fold, we selected one group as the test dataset and the remaining 2 groups as the training dataset. We then calculated the correlation coefficients between predicted and true ribosome densities for the codons in the test datasets.

Table R1. Prediction performance of Riboformer for two replicates from the yeast and worm aging datasets. The model performance was evaluated either by using only one of the two replicates, or by taking the average ribosome densities from both replicates. We performed 3-fold cross-validation tests ($n = 3$), and the Pearson correlation coefficient and Spearman correlation coefficient between the true and predicted ribosome densities are reported (mean \pm SD). This table is labeled as **Supplementary Table3** in the revised manuscript.

Dataset	Metric	Riboformer		
		Rep1	Rep2	Mean
S. cerevisiae (2315 genes)	Pearson	0.928 \pm 0.001	0.913 \pm 0.005	0.936 \pm 0.006
	Spearman	0.808 \pm 0.001	0.816 \pm 0.001	0.862 \pm 0.002

C. elegans	Pearson	0.928 ± 0.004	0.903 ± 0.040	0.938 ± 0.017
(3030 genes)	Spearman	0.732 ± 0.002	0.765 ± 0.001	0.822 ± 0.002

Our analysis revealed that using only one replicate can still yield high correlations between the true and predicted ribosome densities across two organisms (average Pearson correlation is 0.92 for both yeast and worm). Using the average ribosome density resulted in a slight increase in correlations (by 0.02 for both datasets). This suggests that having more replicates could enhance the signal-to-noise ratio and the model's performance. We have revised the manuscript to include these clarifications, and added the details of our computational experiments in **Supplementary Note 2**.

Revised text (Section “Riboformer accurately clarifies ribosome density”):

We also found that Riboformer’s prediction accuracy is robust across replicated experiments while increasing the number of replicates could further enhance the model accuracy (Supplementary Table 3).

4. The examples shown in Figure 1 were related to the technical issues in bacterial ribosome profiling. On the other hand, the application shown in Figure 2 was more related to the biological interpretation of the data. These two themes were totally different but not well introduced in the manuscript. The authors should consider more helpful and careful explanations of these subjects.

We thank the reviewer for this helpful comment. Following the reviewer’s suggestion, we have systematically reorganized the main figures to improve the clarity of our manuscript:

1. In Figure 1, we highlight the model structure, training process and how to apply the trained models to new datasets.
2. Figure 2 demonstrates ability to capture context-dependency of translation dynamics induced by biased experimental protocols. It also showcases the performance of the trained Riboformer model to correct technical bias in unseen datasets.
3. In Figure 3, we analyze an existing dataset using the trained Riboformer model and identify a potential bottleneck in translation elongation.
4. Figure 4 illustrates how Riboformer can be used to identify sequence determinants of ribosome collision.
5. In Figure 5, we used the trained Riboformer model to predict disome formation sites in unseen dataset.

5. In Supplementary Figure 5b and Supplementary Figure 6c, at the disome pause site, there was a complete absence of monosome footprints, which was not normally expected. Could the authors provide more explanation or discussion about these observations?

We thank the reviewer for careful reading of our manuscript. We apologize for the lack of clarity in our original figures. The original Supplementary Fig. 5b shows the normalized

ribosome occupancies of the 40 amino acids (AA) window centered around the disome formation site. In this plot, both monosome and disome footprint reads were normalized, and the reads at each codon was divided by the mean number of reads throughout the 40 AA window. As a result, the normalized occupancy for monosome footprints is around 1, which is shown in the blue line, indicating a constant level of monosome footprints around the disome formation sites. Similarly, in the original Supplementary Fig. 6c, the average ribosome occupancy of young yeast is fluctuated around 1, with a minor peak near the aging-specific pausing sites, in accord with the original work (Stein et al. Nature 2022).

We have revised the figure legend to provide more details on our calculation and revised the y axis labels to highlight the normalized value, as detailed below:

Figure R3. Average ribosome occupancy of the disome profiles around disome peaks (orange) and the average ribosome occupancy in the monosome profiles (blue) are shown, $n = 11,079$ sites. The monosome and the disome profiles are normalized in the 40-codon window by dividing the mean number of reads across the window. This figure is the original **Supplementary Fig. 5b** and the new **Supplementary Fig. 6b**.

Minor points

1. In Supplementary Figure 7f, this EPA codon enrichment result appears to be in apparent contradiction with the results of the original paper (Stein et al. Nature 2022). Explanation regarding this will be helpful for readers.

We thank the reviewer for careful reading of our paper and noticing this. In the original Supplementary Fig. 7 of our work, we compared ribosome profiling dataset from day 1 and 12 and selected all the aging-specific pause sites to conduct SIS analysis. These pause sites were clustered according to their sequence determinants, and we found that Asp is enriched in the P site for several clusters. In the original paper (Stein et al. Nature 2022), the authors reported the peptide motif associated with age comparisons (day 12 vs day 1) in Extended Data Fig. 9c, where Asp also shows the highest enrichment score in the P site. Notably, the peptide motif in Extended Data Fig. 9c is different from Figure 4a in the original paper. In Figure 4a, the authors selected the ribosome pause sites in day 12 ($n = 587$ sites with day 12 pause score >10 in 437 genes).

2. In Supplementary Figure 8, the authors should discuss whether FMRP binding to mRNA and the subsequent translation modulation is beneficial or detrimental to SARS-CoV-2 infection.

We thank the reviewer for this insightful question regarding the role of FMRP binding in SARS-CoV-2 infection. We hypothesize that binding of FMRP to viral mRNA negatively influences viral mRNA translation and subsequently hinder SARS-COV-2 infection. An independent experiment also confirms the therapeutic potential of fragile X syndrome (FXS) drugs for inhibiting SARS-CoV-2 viral reproduction.

(1) Antiviral activity of FMRP in other viruses.

According to Soto-Acosta et al., FMRP binds to the flavivirus RNAs of Zika virus. The depletion of FMRP elevated viral titers and rate of infection by two to five-fold. The antiviral effect of FMRP was further corroborated by immunofluorescence (IF) of infected cells, which suggests that FMRP is a ZIKV restriction factor (Soto-Acosta et al. 2018).

(2) A fragile X syndrome drug inhibits SARS-CoV-2 replication *in vitro*.

G-protein-coupled receptor mGluR5 is a leading drug target for fragile X syndrome that signals through FMRP. Interestingly, a mGluR5 Inhibitor CTEP was reported to reduce viral load in an *in vitro* SARS-CoV-2 assay, implying that FMRP could be a potential restriction factor against SARS-CoV-2 (Westmark et al. 2020).

We have revised the main text to include discussion on the potential role of FMRP on SARS-CoV-2 replication, which is also reproduced in the box below.

Revised text (Section “Discussion”):

Interestingly, antiviral activity of FMRP has been reported for ZIKA virus⁴⁷. In addition, a new study reveals that the SARS-CoV-2 virus load is reduced with the inhibition of mGluR5, a leading drug target for Fragile X syndrome that signals through FMRP⁴⁸.

References:

Soto-Acosta R, Xie X, Shan C, Baker C, Shi P, Rossi S, Garcia-Blanco M, Bradrick S (2018) Fragile X mental retardation protein is a Zika virus restriction factor that is antagonized by subgenomic flaviviral RNA eLife 7:e39023.

Westmark C, Kiso M, Halfmann P, Westmark P, Kawaoka, Y (2020) Repurposing Fragile X Drugs to Inhibit SARS-CoV-2 Viral Reproduction Front. Cell Dev. Biol. 8

3. At line 59, “The” should be “the”.

Thanks. We have rewritten the paragraph.

4. Some of figure citations should be corrected:

At line 110, “Fig 1f” should be Fig. 1e;

At line 118, “Fig 1g” should be Supplementary Fig 2;

At line 114, “Fig. 1h” could not be guessed which figure should be cited here (but definitely

not Fig. 1h);

At line 126, "Fig 1g" should be Fig. 1f and 1g.

We apologize for the mistakes and have corrected our figure legends in the revised manuscript accordingly.

5. At lines 146-147, the authors described Trp codon enrichment in the first 80 codons in low TE genes (Supplementary Figure 4). However, the meaning of these data was not clear. Authors should consider removing these data, or alternatively provide a discussion/explanation of how these data should be interpreted.

We thank the reviewer for this suggestion. We have removed the original Supplementary Figure 4 in the revised manuscript.

Reviewer #2 (Remarks to the Author):

In this manuscript, Shao et al developed Riboformer, a deep learning method to model ribosome protected reads density, and showed a few of its applications. Though deep learning modeling ribosome profiling reads is an interesting topic, the manuscript does not appear to be well written in general. Its overall logic is not clear to readers. It lacks sufficient details and falls short in explaining the results of in silico modeling. Moreover, additional experiment validations are required to support the in silico findings. Overall, the manuscript might not be suitable for publication without plenty of additional work.

We would like to thank the reviewer for the helpful comments.

Major point:

1. The performance of Riboformer would need to be evaluated with independent test data (i.e. AUROC, AUPRC, and the correlation between the modeling density and the true density)

We thank the reviewer for this suggestion. We would like to clarify that for all our analyses, we modeled the ribosome densities and cast the prediction task as a regression problem. As the reviewer suggested, we conducted new computational experiments to evaluate the performance of our model using Pearson and Spearman correlation coefficients between true and predicted ribosome densities with independent test data. The Pearson correlation coefficient is also used by two existing methods: RiboMIMO (Tian et al. 2021) and Riboexp (Hu et al. 2021). Metrics like AUROC and AUPRC, designed for binary classification tasks, are not relevant for our study and are also not reported in RiboMIMO and Riboexp.

To make our results comparable with existing methods, we have followed the previous work (Tian et al. 2021) for model evaluation. Specifically, we reported the average correlation coefficient between the predicted and the true ribosome density across four datasets from three species. For each dataset, we performed 3-fold cross-validation tests to obtain the average performance. In each fold, one subset of the data was held out as test data; the remaining two folds were used as training data. Our results are reported in **Table R2**. We found a high Pearson correlation between the predicted and true ribosome densities for all four datasets (Pearson correlation coefficient ranges from 0.74 to 0.94).

Table R2. Prediction performance of Riboformer in terms of the correlation between true and predicted ribosome densities for all codons in the test dataset. Riboformer model compares ribosome distributions across two datasets and extracts the sequence features driving the difference between them (**Fig. 1b**). The input and output data for corresponding experiments are included in the table. We performed 3-fold cross-validation tests on four datasets across three species (n = 3). The mean \pm SD of the Pearson correlation coefficients and Spearman correlation coefficients are shown. This table is labeled as **Supplementary Table1** in the revised manuscript.

Dataset (gene number)	Input data	Output data	Metric	Riboformer
------------	-------------	--------	------------

E. coli (1005 genes)	Filtering with the Cm-lysis buffer	Flash-freezing with the high-Mg buffer	Pearson	0.891 ± 0.002
			Spearman	0.870 ± 0.001
Yeast (1608 genes)	Monosome	Disome	Pearson	0.744 ± 0.004
			Spearman	0.602 ± 0.006
Yeast (2315 genes)	0 day (young)	4 days (old)	Pearson	0.936 ± 0.006
			Spearman	0.862 ± 0.002
C. elegans (3030 genes)	1 day (young)	12 days (old)	Pearson	0.938 ± 0.017
			Spearman	0.822 ± 0.002

We have revised our manuscript to include more details about our computational methods and metric.

Revised text (Section “Riboformer accurately clarifies ribosome density”):

More specifically, we chose the top 1005 genes with the highest ribosome densities to construct a dataset of 323,688 instances of codons was collected, which was then randomly separated into the training and validation sets. Then we used 3-fold cross-validation tests to evaluate the model performance. In each fold, one subset of the data was held out as test data while the remaining two folds were used as training data. we used the Pearson and Spearman correlation coefficients to measure the correlations between the predicted and true ribosome densities for all codons in the test datasets.

References:

Tian T, Li S, Lang P, Zhao D, Zeng J (2021) Full-length ribosome density prediction by a multi-input and multi-output model. *PLoS Comput Biol* 17(3): e1008842.

Hu H, Liu X, Xiao A, Li Y, Zhang C, Jiang T, Zhao D, Song S, Zeng J (2021) Riboexp: an interpretable reinforcement learning framework for ribosome density modeling, *Brief in Bioinform*, 22(5), bbaa412

2. A comprehensive comparison of different existing methods of modeling ribosome protected reads such as ROSE, RiboMIMO, Riboexp (PMID: 28957655, 33770074, 33479731) would be needed.

We would like to thank reviewer #2 for suggesting existing methods for comparison. RiboMIMO (Tian et al. 2021) and Riboexp (Hu et al. 2021) are two state-of-the-art deep learning-based methods that model ribosome distribution based on sequence features. To fully address this question, we conducted a series of computational experiments to evaluate RiboMIMO, Riboexp and Riboformer on datasets from three species. To make the comparison with existing methods more feasible, we specifically deleted the reference input branch of Riboformer (seq only mode). In this case, all three models take the coding sequence as the input and predict the corresponding ribosome densities. Our results are reported in **Table R3** and **Fig. R3**.

Details for the benchmark experiments are as follows:

Methods:

Following the original publications, we implemented cross-validation tests for performance assessment of Riboexp and RiboMIMO. The default hyperparameters were used. Riboformer was tested in two modes: the seq-only mode which only uses the coding sequence and the full mode which takes two inputs. We used three-fold cross-validation to benchmark the model performance and the mean correlation between true and predicted ribosome densities for codons in the test dataset are reported for all three models.

E. coli dataset:

We used the top 1005 highly expressed genes to benchmark all three methods. In the RiboMIMO paper, 1598 highly expressed genes from the same dataset were used for model testing. In the Riboexp paper, the 500 most highly expressed genes from the same dataset were used. For the full mode of Riboformer, the reference experiment is the ribosome profiles with Cm in the lysis buffer. In our benchmark experiments, the average Pearson correlation of true and predicted ribosome densities was 0.66 for RiboMIMO, which is similar to the original publication (0.69). The average Pearson correlation is 0.64 for Riboexp, while the original paper has reported a Pearson correlation of 0.77 for top 500 highly expressed genes and a correlation of 0.61 for 1375 lower expressed genes. The reason our correlation was 0.13 lower is that we evaluated the model using a larger set of genes, which included those with low sequence coverage. Riboformer's seq only mode generates an average Pearson correlation of 0.81.

Yeast dataset:

We selected the 2315 highly expressed genes from the yeast aging dataset (day 4) to benchmark all three methods. For the full mode of Riboformer, we used the yeast aging (day 0) as the reference input. Both RiboMIMO (0.58) and Riboexp (0.59) had slightly lower Pearson correlation coefficients than the *E. coli* dataset. This trend is also consistent with the original publications. For the seq only mode of the Riboformer model, the average Pearson correlation between the true and predicted ribosome densities is 0.85.

C. elegans dataset:

We selected 3030 highly expressed genes from the aging dataset (day 12) to benchmark all three methods. For the full mode of Riboformer, we used the aging data (day 1) as the reference input. Performance for RiboMIMO and Riboexp on the *C. elegans* dataset was lower compared to the *E. coli* and Yeast datasets, and the Pearson correlations are 0.53 and 0.51 respectively. Riboformer still shows a high correlation between predicted and true ribosome densities (Pearson correlation is 0.88 and 0.94 for the seq-only and full mode).

Table R3. Comparison of prediction performance of Riboformer with that of different baseline methods in terms of the correlation between true and predicted ribosome densities. Following the original publications, we performed 3-fold cross-validation tests for Riboexp (n = 3), and 10-fold cross-validation tests for RiboMIMO (n = 10). Riboformer model was evaluated in two modes: “seq only mode” which uses the coding sequence as the only input, and the “full mode” which takes two inputs. We used 3-fold cross-validation tests for Riboformer (n = 3). All models were tested on an identical set of highly expressed genes for each species. The mean \pm SD of the Pearson correlation coefficients and Spearman

correlation coefficients are shown. This table is labeled as **Supplementary Table4** in the revised manuscript.

Dataset (gene numbers)	Metric	Methods			
		Riboexp	RiboMIMO	Riboformer (seq only)	Riboformer (full mode)
E. coli (1005 genes)	Pearson	0.642 ± 0.005	0.657 ± 0.024	0.814 ± 0.013	0.891 ± 0.002
	Spearman	0.656 ± 0.002	0.683 ± 0.013	0.833 ± 0.002	0.870 ± 0.001
Yeast (2315 genes)	Pearson	0.587 ± 0.002	0.581 ± 0.012	0.846 ± 0.008	0.936 ± 0.006
	Spearman	0.489 ± 0.010	0.491 ± 0.009	0.783 ± 0.002	0.862 ± 0.002
C. elegans (3030 genes)	Pearson	0.511 ± 0.026	0.534 ± 0.009	0.877 ± 0.019	0.938 ± 0.017
	Spearman	0.467 ± 0.010	0.497 ± 0.009	0.773 ± 0.002	0.822 ± 0.002

ROSE (Zhang et al. 2017) is not designed for and does not support prediction of continuous ribosome density. It focuses on identification of ribosome stalling events using the sequence features. In the original publication, the performance of ROSE is measured using the area under the corresponding ROC curve (AUROC) score, which is also specific to binary classification tasks. Therefore, we believe it is out of scope for this manuscript to adapt and test ROSE on the same datasets by implementing the necessary modifications to properly perform this comparison.

Although we were unable to properly benchmark ROSE, we believe it is important to acknowledge this effort in our manuscript. Therefore, we have revised the manuscript to briefly discuss this method.

Finally, we want to emphasize that Riboformer stands in its own ability to model relative changes of ribosome occupancy, correct for the experimental bias, and predict ribosome density in new experiments. Although we have benchmarked the seq only mode of Riboformer which takes only the coding sequence as the input, its application is not the focus of this work.

We have added these results in the revised manuscript. We also included a supplementary note about the details of the benchmark experiments (**Supplementary Note 1**).

Revised text (Section “Introduction”):

Convolutional neural network (CNN) has been implemented to predict ribosome stalling sites in both yeast and human cells¹⁴, outcompeting the conventional methods. More recently, deep learning paradigms such as RiboMIMO¹⁵ and Riboexp¹⁶ were developed to reconstruct the ribosome density distributions based on the coding sequence (CDS).

Revised text (Section “Riboformer accurately clarifies ribosome density”):

Finally, we systematically compared the performance of Riboformer with other existing deep learning-based models including RiboMIMO and Riboexp (Supplementary Fig. 3 and Supplementary Table 4).

Figure R3. Comparison of prediction performance of Riboformer with that of different baseline methods across three different species. Riboformer model was tested in two modes: “seq only mode” which uses the coding sequence as the only input, and the “full mode” which takes two inputs including the coding sequence and the reference input. Each dot represents one codon from the independent test datasets. Results from one-fold of the cross-validation tests are shown. The results from all folds are reported in **Table R3**. r and ρ are Pearson and Spearman correlation coefficients between the true and predicted ribosome density for all codons in the test datasets. This figure is labeled as **Supplementary Fig. 3** in the revised manuscript.

References:

Zhang S, Hu H, Zhou J, He X, Jiang T, Zeng J (2017) Analysis of Ribosome Stalling and Translation Elongation Dynamics by Deep Learning. *Cell Systems* 5(3) P212-220.E6

Tian T, Li S, Lang P, Zhao D, Zeng J (2021) Full-length ribosome density prediction by a multi-input and multi-output model. *PLoS Comput Biol* 17(3): e1008842.

Hu H, Liu X, Xiao A, Li Y, Zhang C, Jiang T, Zhao D, Song S, Zeng J (2021) Riboexp: an interpretable reinforcement learning framework for ribosome density modeling, *Brief in Bioinform*, 22(5), bbaa412

3. In Fig2, authors identified sequence determinants of ribosome pausing as a new finding. The related experimental validations should be provided to support these modeling findings. The same causality-validation mindset might be applicable to a few other in silico modeling findings in the manuscript.

We thank the reviewer for raising this important point. Our model findings agree with existing knowledge on ribosome elongation modulation, yet Riboformer generates actionable hypotheses for the regulation of ribosome stalling events. More specifically, SIS analysis identifies ribosome stalling sites that could be mediated by proline enriched motifs, mRNA secondary structure, and polybasic stretches such as consecutive Lys codons. It further implies the potential of fragile X syndrome (FXS) drugs for inhibiting SARS-CoV-2 viral reproduction, which is validated in an independent experiment.

(1). Riboformer captures the role of Proline enriched motif in ribosome stalling.

Proline has been shown to slow down translation elongation across different organisms (O'Connor et al. 2016). In the disome experiment, Riboformer provides an in-depth classification of ribosome stalling sites based on the SIS profiles. We found that for cluster 7-10, the ribosome decoding sites have low SIS, indicating that they are the sequence determinant of ribosome stalling (**Fig. 4**). Intriguingly, we also found these sequences are highly enriched in Proline.

(2). Riboformer reveals ribosome stalling events that are mediated by mRNA folding.

Although it is widely known that mRNA secondary structure could mediate ribosome pausing (Andreev et al. 2017), it is still not clear which ribosome pausing sites are more prone to the effect of mRNA folding. In our analysis of all the clusters, we observed that cluster 4 had a low SIS value in its downstream sequence, potentially due to mRNA folding (**Fig. R1a**). Indeed, we found that the sequences from this cluster show the highest structure score among all the clusters (**Fig. R1c**).

(3). Riboformer identifies polybasic stretches that could induce ribosome stalling.

Previous works have demonstrated that consecutive Lys codons (polybasic region) could be potential sites for ribosome collision (Stein et al. 2022). In the PWP1 gene, the monosome profile doesn't show peaks in the polybasic region, while the disome profile shows isolated

peaks. Interestingly, our algorithm identified consecutive Lys codons as the sequence determinant of disome peaks (**Fig. R2**).

Figure R2. SIS analysis identifies consecutive Lys codons as the sequence determinant of disome peaks. Monosome (gray, top) and disome (blue, middle) profiles of the PWP1 gene in WT yeast cells are shown. The SIS profile (bottom) around the disome peaks is shown and the region with consecutive Lys codons is shaded in light gray. This figure is labeled as **Supplementary Figure 8** in the revised manuscript.

(4). Riboformer implies the potential value of fragile X syndrome drugs for inhibiting SARS-CoV-2 viral reproduction.

By using SIS analysis, we have identified the potential role of FMRP in the translation modulation of SARS-CoV-2 replication. This finding is consistent with prior research on viruses such as ZIKA, where studies have shown that FMRP binds to the flavivirus RNAs of the Zika virus and inhibits viral replication (Soto-Acosta et al. 2018). In addition, independent experiments have proved the ability of a fragile X syndrome drug (CTEP, an inhibitor of mGluR5 that signals through FMRP) to reduce viral load in an in vitro SARS-CoV-2 assay (Westmark et al. 2020).

Due to the nature and scope of the paper, we are not equipped to perform additional experimental validations. We acknowledge that experimental validations could further strengthen our results and interpretations, and we recommend such follow-up experiments as a valuable avenue for future research. The modifications to our manuscript are reproduced in the box below.

Revised text (Section “Riboformer identifies sequence determinants of ribosome collisions”):

Cluster 4 and 5 show distinct SIS profiles from the population average, with Trp and Lys codons enriched at the ribosomal A site, respectively (Supplementary Fig. 7). We found pausing sites from cluster 4 are more likely to be affected by the mRNA secondary structure

of the downstream sequences (Supplementary Fig. 7c). In addition, previous works have demonstrated that consecutive Lys codons (polybasic region) could be potential sites for ribosome collision^{7,36}. Riboformer further identifies consecutive Lys codons as the sequence determinant of disome peaks in the PWP1 gene (Supplementary Fig. 8).

Revised text (Section “Discussion”):

Interestingly, antiviral activity of FMRP has been reported for ZIKA virus⁴⁷. In addition, a new study reveals that the SARS-CoV-2 virus load is reduced with the inhibition of mGluR5, a leading drug target for Fragile X syndrome that signals through FMRP⁴⁸.

References:

O’Connor, P. B. F., Andreev, D. E. & Baranov, P. V. Comparative survey of the relative impact of mRNA features on local ribosome profiling read density. *Nat Commun* 7, 12915 (2016).

Andreev, D. E. et al. Insights into the mechanisms of eukaryotic translation gained with ribosome profiling. *Nucleic Acids Res* 45, 513–526 (2017).

Stein, K. C., Morales-Polanco, F., van der Lienden, J., Rainbolt, T. K. & Frydman, J. Ageing exacerbates ribosome pausing to disrupt cotranslational proteostasis. *Nature* 601, 637–642 (2022).

Soto-Acosta, R. et al. Fragile X mental retardation protein is a Zika virus restriction factor that is antagonized by subgenomic flaviviral RNA. *Elife* 7, e39023 (2018).

Westmark, C. J., Kiso, M., Halfmann, P., Westmark, P. R. & Kawaoka, Y. Repurposing Fragile X Drugs to Inhibit SARS-CoV-2 Viral Reproduction. *Front Cell Dev Biol* 8, (2020).

4. The manuscript should be re-written clearly to show its full logic and details to readers.

We thank the reviewer for this suggestion. To fully address this issue, we have systematically revised the manuscript to improve its clarity:

- (1) We have re-written the introduction to highlight existing methods and the current challenges.
- (2) We have added a paragraph in the discussion part about the limitations of our work and potential solutions.
- (3) We have reorganized all the main figures to improve the logic of the paper (**new Fig. 1-5**). In the revised figure 1, we highlight the model structure, training process and how to apply the trained models to new datasets; new Figure 2 demonstrates Riboformer’s ability to capture context-dependency of translation dynamics induced by biased experimental protocols; in the new Figure 3, we analyze an existing dataset using the trained Riboformer model and identify a potential bottleneck in translation elongation; new Figure 4 illustrates how Riboformer can be used to

identify sequence determinants of ribosome collision; in the new Figure 5, we used the trained Riboformer model to predict disome formation sites in unseen dataset.

- (4) We have added two supplementary notes to provide more details about the computational experiments (**Supplementary Note 1 and 2**), Supplementary Note 1 details the benchmarking experiments for the comparison of Riboformer with RiboMIMO and Riboexp. Supplementary Note 2 details the impact of input data quality on the Riboformer's performance. We also extended the method section of the manuscript to provide more details about the model structure.

Reviewer #3 (Remarks to the Author):

In this manuscript Shao et al describe software called Riboformer for the analysis of ribosome profiling data. The software is based on the use of convolutional neural networks (CNN). In essence Riboformer can compare ribosome profiling densities for a set of genes from two datasets (say A and B), learn what sequence features (within 120 nt window) are responsible for the difference between A and B and then use this knowledge to predict how ribosome densities would look in B given data from A. This is a highly useful functionality as demonstrated by the authors. Previously Buskirk and colleagues (ref 16) have developed an experimental approach for obtaining ribosome profiling to minimise the distortions in ribosome density introduced by experimental procedures (e.g. drug chloramphenicol treatment, culture filtering and scrapping). This procedure is laborious and is hard to implement. By using Riboformer one could use an easier protocol that produce less accurate data and use Riboformer to predict how the data would look like if they were obtained with the “unbiased” protocol. The authors also successfully demonstrated the utility of Proetofomer as data transformation tool to predict disome densities from monosome profiling and to predict pauses expected in aged cells. Furthermore, the authors came up with a clever approach to look into the “black box” of Riboformer’s CNN. For this they performed an in silico mutagenesis and explored mutations that influence pausing, thus figuring out what sequence features are expected to cause pauses. So overall, Riboformer is a very useful addition into a set of computational tools for ribosome profiling data analysis, I can clearly see how it can be used by others and for additional applications. However, I have several critical comments and suggestions for improvements.

We would like to thank the reviewer for the insightful comments, and we appreciate the reviewer’s positive feedback.

General criticism

1. First of all, we were not able to install and run software, even though tried several different computers through conda on both in Ubuntu 22LTS and Windows 10. Some error logs are at the end of this report. The authors should test Riboformer on several different platforms and try to maximise its compatibility. Perhaps it makes sense to explicitly describe environment requirements. To make the tool useful it should be reasonably easy to install and run.

We wish to thank the reviewer for carefully testing Riboformer and bringing the installation issues to our attention. We apologize for the inconvenience this has caused. We have since addressed the reported problems and updated the installation process. We have provided the updated yaml file and the software can be easily installed via Conda:

```
conda env create -f environment.yaml
```

We have confirmed its compatibility across multiple platforms, including Ubuntu 22/24, Windows 11 (64 bit), and Linux (Debian).

2. Although the manuscript is written in good language it describes Riboformer's capabilities in terms that are too generic. The limitations of Riboformer also not clearly stated, as a result it was not easy to understand what it specifically does. I suggest trying to use more specific language. It will make it easier for user and readers to understand what Riboformer does and what should it be used for. Several specific points below are illustrations.

Specific critical points and suggestions:

3. The abstract states that Riboformer "corrects experimental artefacts in previously unseen datasets". Also on lines 133-134 "Riboformer is a predictive framework that can be used to standardize a wide range of ribosome profiling measurements, reducing experimental noise while remaining true to the underlying biological signal of interest."

Strictly speaking Riboformer alone does not do this. If we have an unbiased dataset Proetofomer can learn how to remove the biases in the biased datasets. So the performance of Riboformer is critically dependent on the existence of the improved experimental datasets. This point is not communicated clearly in the manuscript. Even figure 1 gives an impression that Riboformer takes as input ribosome profiling dataset and a sequence and somehow magically produces unbiased dataset. But this is a false impression because Riboformer can do it only if an "unbiased" dataset is given for training (hence it should be shown as a part of input). In the specific examples used by the authors, Riboformer is used to convert data obtained with Chloramphenicol to what would be expected if the data were obtained with blocking the ribosomes with high Mg concentration. If high Mg concentration was perturbing ribosome locations, Riboformer would introduce (rather than remove) the experimental biases.

We thank the reviewer for this helpful comment. We apologize for the potential confusion. Following the reviewer's suggestions, we have revised our manuscript to improve its clarity. In addition, we have reworked **Fig. 1** to emphasize the model training process:

Figure R4. Overview of the Riboformer pipeline. a, applications of the Riboformer pipeline. **b**, schematic illustration of the neural network structure of Riboformer. This figure is labeled as **Fig. 1** in the revised manuscript.

Revised text (Section “Abstract”):

When trained on an unbiased dataset, Riboformer can correct experimental artifacts in previously unseen datasets.

Revised text (Section “Riboformer corrects experimental bias in unseen data”):

Collectively, these results demonstrate that once trained on the unbiased datasets, the Riboformer model can be used to standardize...

It would be helpful if the authors could also demonstrate RiboFormer performance to remove biases arising from drug treatment artefacts using the biased (chx) and unbiased (chx/tig) data from Wu et al 2019 doi: 10.1016/j.molcel.2018.12.009.

As the reviewer suggested, we benchmarked the performance of Riboformer using the Wu et al. dataset. We focused on three different read counts: the total read counts (21 nt RPFs + 28 nt RPFs), the short reads (21 nt RPFs), and the long reads (28 nt RPFs). According to the original publication, 21 nt RPFs are derived from ribosomes with an open A site in vivo, while 28 nt RPFs correspond to ribosomes with an occupied A site. 21 nt RPFs from samples prepared with CHX/TIG showed the strongest correlation with tRNA abundance (unbiased data).

To make our trained models more applicable to the other datasets, we did model training in three ways:

- (1) Predict unbiased (chx/tig) 21nt reads from biased (chx) 21nt reads.
- (2) predicting unbiased (chx/tig) 21 nt reads from biased (chx) all reads.
- (3) predicting unbiased (chx/tig) 28 nt reads from biased (chx) all reads.

We used three-fold cross validation tests to evaluate the model performance in all three cases. The correlations of the predicted and true ribosome densities in the target condition are reported in **Table R4**. We observed a high correlation between the predicted and the true ribosome densities. The Pearson correlation coefficient between true and predicted ribosome densities ranges from 0.68 to 0.80. We also include this result in the revised manuscript.

Table R4. Prediction performance of Riboformer for correcting experimental bias in yeast. Wu et al. dataset was used to train the Riboformer models. We generated three different datasets: ribosome densities for 21 nt ribosome-protected mRNA fragments (RPFs), 28 nt RPFs, and 21 nt + 28 nt RPFs, under two conditions: cycloheximide (CHX) treatment and combined cycloheximide (CHX)/tigecycline (TIG) treatment. Read counts from the CHX treatment were used to predict open 40S A sites (21 nt), or occupied 40S A sites (28nt) under CHX/TIG treatment. 21 nt RPFs from samples prepared with CHX/TIG showed the strongest correlation with tRNA abundance (unbiased data). We performed 3-fold cross-validation tests ($n = 3$), and the mean \pm SD of the Pearson correlation coefficients and Spearman correlation coefficients are shown. This table is labeled as **Supplementary Table 5** in the revised manuscript.

Dataset (Description)	Metric	Riboformer
Wu dataset (CHX 21nt to CHX/TIG 21 nt)	Pearson	0.731 \pm 0.007
	Spearman	0.700 \pm 0.002
Wu dataset (CHX all to CHX/TIG 21 nt)	Pearson	0.680 \pm 0.009
	Spearman	0.689 \pm 0.005
Wu dataset (CHX all to CHX/TIG 28 nt)	Pearson	0.798 \pm 0.018
	Spearman	0.816 \pm 0.001

Revised text (Section “Discussion”):

We further demonstrated that Riboformer can be trained to correct technical bias induced by CHX treatment in yeast⁴⁴, indicating Riboformer's broad applicability to existing ribosome profiling datasets (Supplementary Table 5).

4. Another abstract statement Riboformer “uncovers a bottleneck in protein synthesis”, also the title of one of the subsections “Riboformer allows identification of limiting steps in protein synthesis”. This statement is too generic and obscure, many people would think of translation initiation and availability of certain initiation factors as bottlenecks in protein synthesis. Here, however, the authors reported finding Trp deficiency that effects progression of the ribosomes during the elongation state. To what extent this deficiency effects the rate of protein synthesis is not yet clear. A more specific description of the finding would avoid potential misinterpretation of the claim regarding the bottlenecks.

We would like to thank the reviewer for this insightful feedback. We recognize the ambiguity in our initial claim. Since the translation initiation rate in *E. coli* is much faster than higher organisms like yeast (Gorochofski et al. 2019; Shah et al. 2013), we hypothesize that Trp deficiency could be a potential rate limiting step for protein synthesis in burdened cells. However, as the reviewer noted, it is still hard to determine the exact impact of decoding of Trp codons on overall protein synthesis. This would require absolute quantification of translation initiation rate and elongation rates, which are not provided in the original publication. We have clarified this in the revised abstract and main text.

Revised text (Section “Abstract”):

...which reveals subtle differences in synonymous codon translation and uncovers a bottleneck in translation elongation.

Revised text (Section “Riboformer allows identification of a limiting step in translation elongation”):

Thus, our results indicate that slow decoding of Trp codons could affect translation elongation. Since translation initiation is much faster in *E. coli* than in eukaryotic cells²⁹, our findings suggest the potential role of Trp deficiency as a rate limiting step in protein synthesis.

References:

Gorochofski T, Chelysheva I, Eriksen M, Nair P, Pedersen S, Ignatova Z (2019) Absolute quantification of translational regulation and burden using combined sequencing approaches. *Molecular Systems Biology* 15:e8719

Shah P, Ding Y, Niemczyk M, Kudla G, Plotkin J (2013) Rate-Limiting Steps in Yeast Protein Translation, 153(7) P1589-1601.

5. Some descriptions are vague and as a result appear contradictory, i.e. “that detects long-range dependencies in the regulation of elongation (Fig. 1a).” and “More specifically, our approach assumes that the relative change in ribosome occupancy is primarily determined by the surrounding sequences.” So is it long distance or surrounding?

We apologize for this confusion. As the reviewer pointed out, Riboformer uses the surrounding sequence (120 nt) which is not long-range. Following the reviewer’s suggestion, we have deleted this confusing description in the revised manuscript to improve clarity.

In practice, the authors explore a window of 120 nucleotides centred at the A-site of the footprint generating ribosome. I think the use of “long-range dependency” here may be misleading as some may think of much longer intervals. The choice of 120 nucleotides sounds reasonable intuitively, it takes into account the sequence encoding the polypeptide inside the ribosome that may influence its progression as well as downstream sequence responsible for interfering RNA structures and effects of downstream ribosomes. However, this appears as an arbitrary choice. Did the authors play with the length of the window? It would be nice to explore how the size of the window affect the accuracy of transformations.

We thank the reviewer for this helpful suggestion. To fully address this question, we conducted new computational experiments using varying window sizes, specifically 30 nt, 60 nt, 240 nt, and our initial choice of 120 nt. We used three-fold cross-validation tests to evaluate model performance, and the Pearson and Spearman correlation coefficients between true and predicted ribosome densities are reported in **Fig. R5**.

Figure R5. Prediction performance of Riboformer in relation to the window size of input sequence. Riboformer was evaluated using 3-fold cross-validation tests on the Mohammad et al. dataset ($n = 3$). Pearson correlation coefficients (left) and Spearman correlation coefficients (right) between the true and predicted ribosome densities are shown. The error bars represent the standard deviation. This figure is labeled as **Supplementary Fig. 1** in the revised manuscript.

We found that Pearson’s correlation coefficients are consistent across window sizes of 60 nt, 120 nt and 240 nt. However, for Spearman’s correlation coefficient, model performance increases with the window size, meaning that capturing a longer range of dependency could further improve the model performance. However, we would like to note that (1) the

performance gain is diminishing from 120 nt to 240 nt; (2) for the transformer architecture we used in the Riboformer model, the computational complexity goes quadratic with the window size (context length). Taking these observations into account, we posit that a 120 nt window offers an optimal balance between achieving model accuracy and maintaining manageable computational complexity. We have incorporated this result in our manuscript, which is also reproduced in the box below.

Revised text (Section “Riboformer accurately clarifies ribosome density”):

By varying the window sizes of the input sequence, we observed that the model performance increases with window size (Supplementary Fig. 1). However, the improvement becomes marginal when the window size exceeds 120 nt.

6. The data used for training and testing were processed to include only highly covered transcripts. Although this approach improves signal-to-noise ratio, it also may potentially introduce biases specific to highly expressed transcripts. I wish the authors explored this a little more. How well RiboFormer works on predicting densities at the lowly expressed transcripts? How specific coverage thresholds influence its accuracy?

We thank the reviewer for this great suggestion on evaluating the model’s performance with respect to sequence coverage and potential bias in model training. To fully address these questions, we conducted a series of new experiments. Firstly, we found that Riboformer trained on highly expressed genes can effectively predict ribosome density in lowly expressed genes. In addition, we varied the sequence coverage threshold and evaluated the model performance using cross-validation tests. To gain more insight into our parameter choice, we further compared the threshold we chose with two state-of-the-art methods: RiboMIMO (Tian et al. 2021) and Riboexp (Hu et al. 2021), since we have used the exact same dataset for model training and evaluation (response to reviewer #2: Q2).

Riboformer performance is robust on lowly expressed genes.

We conducted new experiments to ensure that the prediction performance of our model can be generalized to lowly expressed genes. We trained the Riboformer model using the top 25% of highly expressed genes from the *E. coli* dataset. The model performance was then tested on 402 lower expressed genes that are not in the training dataset and fall in the 25th to 35th percentile for expression. On average, 69% of codons in these lowly expressed genes have zero read counts. This training and testing process was repeated three times, and the average performance in terms of the correlation between true and predicted ribosome densities for all codons in the test dataset is reported. In the meantime, we trained and evaluated the baseline models by using only the lowly expressed genes. We found that the model trained on the highly expressed genes generated a Pearson correlation that was only 0.03 less than the baseline models (**Table R5**). This suggests that even when trained on highly expressed genes, the Riboformer model effectively predicts ribosome density for genes with lower expression.

Table R5. Prediction performance of Riboformer on lowly expressed genes. We selected a subset of 402 genes in the Mohammad et al. dataset with low ribosome densities (average ribosome density across coding sequence < 0.09). We performed 3-fold cross-validation tests ($n = 3$) using this dataset to generate baseline performance. In addition, we trained the Riboformer model on the high-density genes and tested the model performance on these lowly expressed genes three times ($n = 3$). The mean \pm SD of the Pearson correlation coefficients and Spearman correlation coefficients are shown. This table is labeled as **Supplementary Table 2** in the revised manuscript.

Model description	Metric	Riboformer
Three-fold cross-validation on lowly expressed genes (402 genes)	Pearson	0.566 ± 0.008
	Spearman	0.542 ± 0.006
Training on highly expressed genes (1005 genes) and testing on lowly expressed genes (402 genes)	Pearson	0.535 ± 0.003
	Spearman	0.515 ± 0.003

Riboformer’s performance is robust to specific coverage thresholds.

We separated the genes according to their sequence coverage into 4 groups (from high to low): 80% to 100%, 60% to 80%, 40% to 60% and 20% to 40%, and used 3-fold cross validation to test Riboformer’s performance on these 4 datasets. Here the sequence coverage is defined as the proportion of codons with non-zero read counts across the gene coding regions. It is worth noting that genes with high averaged ribosome density tend to have increased sequence coverage (**Fig. R6a**). We found that our model performs better for transcripts with high coverage (**Fig. R6 b&c**), in accord with previous work (Tian et al. 2021). For sequence coverage from 40% to 60% we achieved a Pearson correlation of 0.62 between true and predicted ribosome densities. Whereas for very poorly expressed transcripts (sequence coverage $< 40\%$), we still achieve a Pearson correlation of 0.53 between true and predicted ribosome densities (Figure R6).

Figure R6. Prediction performance of Riboformer in relation to the sequence coverage of the input data. The sequence coverage is defined as the proportion of codons with non-zero read counts across the gene coding regions. **a**, average ribosome densities for the genes with different sequence coverage. Number of genes for each group is shown on the top.

Prediction performance of Riboformer for genes with different sequence coverages in terms of the Pearson correlation coefficient (**b**) and Spearman correlation coefficient (**c**) between true and predicted ribosome densities. The model was evaluated using 3-fold cross-validation tests on the Mohammad et al. dataset (n = 3). The error bars represent the standard deviation. This figure is labeled as **Supplementary Fig. 2** in the revised manuscript.

Our model threshold is comparable with previous approaches.

In our benchmarking experiments using the *E. coli* dataset, we chose the top 1005 genes for our modeling training and evaluation. Two existing computational methods (Riboexp and RiboMIMO) have used the same ribosome profiling dataset for model evaluation. In the Riboexp paper, the top 500 highly expressed genes were used for model training. In the RiboMIMO paper, the number of genes they used for model training and evaluation after filtering is 1598, which is a larger number of genes than we used. This difference in gene count is partly because we excluded all genes shorter than 200 nt.

Overall, our computational experiments above dissect how the sequence coverage affects the performance of Riboformer and indicate that its prediction power is not biased towards highly expressed transcripts. We have added the above results into our revised manuscript and also added a new **Supplementary Note 2**.

Revised text (Section “Riboformer accurately clarifies ribosome density”):

We found the model performs better for highly expressed genes due to the high signal to noise ratio (Supplementary Fig. 2). Thus, we assessed potential biases in model performance when trained on genes with high ribosome density. Interestingly, models trained on these genes could effectively predict ribosome density for lowly expressed genes (Supplementary Table 2).

References:

Tian T, Li S, Lang P, Zhao D, Zeng J (2021) Full-length ribosome density prediction by a multi-input and multi-output model. *PLoS Comput Biol* 17(3): e1008842.

Hu H, Liu X, Xiao A, Li Y, Zhang C, Jiang T, Zhao D, Song S, Zeng J (2021) Riboexp: an interpretable reinforcement learning framework for ribosome density modeling, *Brief in Bioinform*, 22(5), bbaa412

The current threshold seems a bit odd, for prokaryotes it is 0.5 reads per nt and for eukaryotes 5 reads per nt. I would have thought it should be the way around since prokaryotic riboseq data usually exhibit higher coverage due to the small size of their genomes.

We apologize for the confusion, and we appreciate the reviewer’s careful observation. We believe these issues arose from different normalization methods used in the processed ribosome profiles. To prevent confusion for users of our software, we have added a new functionality in the data processing function to select the top expressing genes based on a

specified percentile of their average ribosome density. Users can now determine the number of genes to analyze based on quality of the dataset. The threshold could be implemented in the data processing step as follows:

```
python data_processing.py -th=25 # choose the top 25% expressed transcripts
```

We have also improved the online documentation about the data processing step. (<https://github.com/lingxusb/Riboformer#training-riboformer-on-new-dataset>)

7. I think the authors should explicitly describe the limitations of the software. In my opinion these are (i) Reliance on unbiased or condition-specific datasets. (ii) Limited-range dependencies. It is reasonable to assume that certain ribosome behaviours such as ribosome queuing would depend not only the strength of ribosome stalling, but also on the rate with which ribosome initiate on mRNA. That, however, is excluded from the modelling. (iii) The modelling is limited to middle parts of coding regions, excluding regions upstream and downstream and therefore it is unclear whether the transformation would give reasonable results in those regions. (iv) Like with some other previous approaches such as RUST, the model likely captures general factors effecting ribosome dwell times on all or most mRNAs and as a result it may fail to work adequately on special and unique situations, such as ribosomal frameshifting, exceptional stalling sites like, etc. This could be potentially explored, but may not be easy, as one first needs to identify a pair of datasets where experimental or biological conditions effect these rare events.

We would like to thank the reviewer for raising these important points. We agree with the reviewer on these limitations of our work. We have incorporated more discussions in our manuscript, which are also reproduced in the box below.

Revised text (Section "Discussion"):

The Riboformer framework is not without its limitations. Firstly, it relies on existing datasets for training. With the development of techniques for unbiased measurement of translational landscape, we envision that new Riboformer models can be further trained to improve the analysis of biased datasets. In addition, like many existing methods, Riboformer doesn't consider the translation initiation and termination, both of which can affect ribosome queuing along the transcript. Consequently, our model excludes the first and last ten codons in the gene coding region in the downstream analysis. This could be addressed in future work through systematic quantification and modeling of translation initiation and elongation rates. Finally, the Riboformer model is not designed to handle rare events like ribosomal frameshifting, due to the limited number of training samples. To tackle these specific situations, transfer learning approaches could be explored, which allows for initial training on one task and subsequent fine-tuning across various contexts.

8. It would be good if the authors provide more explicit detail on what data need to be provided for Riboformer. Wig files are required, but what should be used as coordinates? Presumably offsetted locations of A-sites, but this is not stated. How the data should be

processed? It is likely that offsetting may affect the outcomes (i.e. read length dependent offsetting vs uniform offsetting), for a novice users some instructions could be very useful in this regard and will help avoid frustration.

We sincerely apologize for the confusion. As the reviewer mentioned, our algorithm requires processed ribosome footprint counts (wig file) as input. We have systematically improved the online documents about the required files (<https://github.com/lingxusb/Riboformer#training-riboformer-on-new-dataset>), and provided example wig files. For example, the wig files for the *E. coli* dataset are the mapped 3'-end reads. We then applied uniform offsetting in the data preprocessing step, following the original publications. The offset (-14 nt for P site) is one of the input parameters to the function "data_processing" (<https://github.com/lingxusb/Riboformer#training-riboformer-on-new-dataset>).

Aside from the wig files, genomic sequence (fasta file) and gene annotations (gff3 file) for the reference genome are also required. We also provided sample data in our github repository:

https://github.com/lingxusb/Riboformer/tree/main/datasets/GSE119104_Mg_buffer, and https://github.com/lingxusb/Riboformer/tree/main/datasets/GSE139036_disome.

9. Pearson coefficients are used as a measure of correlations, but those are very sensitive to outliers and unclusion/omission of a few datapoints may significantly alter the correlations. For the prudence it may be helpful to include Spearman correlations.

We would like to thank the reviewer for this helpful suggestion. We have reported Spearman correlation coefficients for all our computational results including the new **Supplementary Table 1-5** and **Supplementary Fig. 3**.

10. Code for most subfigures is not available, but some are stated in reproducibility section of pipeline, mention this explicitly in Code availability section.

We thank reviewer for raising this point. We have mentioned the availability of the codes for subfigures in the code availability section.

Revised text (Section "Code Availability"):

Codes for the Riboformer pipeline are available from GitHub (<https://github.com/lingxusb/Riboformer>). Codes for reproducing the figures including Fig. 2d, Fig. 2e, Fig. 4b, Fig. 4c, Fig. 4e, Supplementary Fig. 4 and Supplementary Fig. 5, are available from GitHub (<https://github.com/lingxusb/Riboformer/tree/main/reproducibility>).

11. Sup. Figure 6c text: spelling error: . n = 6,347 sites.

Response: Thanks. We have fixed this typo in our revised manuscript.

Error log:

- Installation failed through conda (ubuntu 22), I had to remove line:
“- json=0.0.1=0” in yml file for it to work.
- Program fails to run using example data:
cd Riboformer/Riboformer
python3 transfer.py -i ../datasets/GSE139036_disome -m ../models/yeast_disome
Error:
Data Loading: Ofile [00:00, ?file/s]
Traceback (most recent call last):
File "/home/roler/Desktop/forks/Riboformer/Riboformer/transfer.py", line 86, in <module>
main()
File "/home/roler/Desktop/forks/Riboformer/Riboformer/transfer.py", line 65, in main
x_c[:, :40] = x_c[:, 0:40] / 100 - 5
TypeError: 'NoneType' object is not subscriptable

We wish to thank the reviewer for carefully testing our software. We apologize for these issues. The installation issue has been fixed (response to Q1). We have improved the online documentation to include sample data for the “transfer” function (<https://github.com/lingxusb/Riboformer/tree/main#applying-trained-riboformer-model-on-new-dataset>). We have also updated the code to better report errors.

We would also like to emphasize our commitment to the continuous improvement and maintenance of Riboformer. We genuinely welcome and encourage the research community to report any errors or issues they encounter, and we're open to suggestions for new functionalities that could enhance the tool further.

Pasha Baranov

REVIEWER COMMENTS

Reviewer #1 (Remarks to the Author):

The authors have addressed all the concerns that were raised by this reviewer. This reviewer supported the publications in Nature Communications.

Additional minor comments:

(This reviewer would say that some of the points below should be commented on in the first round of peer review and thus may not be fair to be added in this second round. However, these points may be worth considering to improve this manuscript.)

1. The manuscript defined the window 120 nt or 20 + 20 codons. They are the same but may be helpful to keep consistency throughout the manuscript for easy understanding for readers.
2. On page 6 line 160, it is worth adding the conclusion sentence for the corresponding paragraph.
3. On page 7 lines 197-198 "Since translation initiation is..."; this sentence does not sound logical.
4. TE should be defined properly (footprints normalized by RNA-seq).
5. On page 7 lines 205-206; "Ribosome stalling in the Tryp codon-rich..."; this sentence and the cited figure may not look match well.
6. On page 8 line 236, "tRNA binding sites" should be clarified as more concretely (A, P, or E site)
7. Fig. 4d was not cited in the manuscript.
8. For Supplementary Fig. 7a, Cluster 5, ribosome stalling on R-X-K motif was reported previously (DOI: 10.1016/j.celrep.2020.107610, DOI: 10.1016/j.celrep.2016.07.018). These should be cited in the text.
9. On page 9 line 263-264, Pro codon was enriched in A site in Supplementary Fig. 9e-f (for e, 0 position was guessed as A site), but the sentence highlight E and P sites. This point was a contradiction. Similarly, Fig. 4d showed Pro enrichment at +1 site (one codon downstream from A site) as well. This was not reflected in the logo in Fig. 4e.
10. On page 10 line 291-293, this reviewer did not agree on the description. First, CHX/TIG was not pre-treated in cells in the paper but rather added in lysis buffer. Second, the difference between long and short footprints was not considered as a "bias". They simply represent whether A-site was filled with tRNA or not. More precise writing is definitely required.
11. Regarding Fig. 5c, if this reviewer's understanding is correct, the disome peak at position 0 should be "disome", ~-9 position should be trisome (disome + monosome), and ~-18 position should be tetrasome (disome + monosome + monosome). The cartoon inside may not correctly present the data.
12. Supplemental note 1 and 2 were not cited in the manuscript.

Reviewer #2 (Remarks to the Author):

We appreciate the authors in taking time to address our concerns in the 1st round of review. Though

some improvements, there are several critical issues to be settled before it becomes a manuscript suitable for publication:

1) Using deep learning to modeling ribosome profiling is an revisited approach with a few existing methods already in the field: besides the ROSE, RiboMIMO and Riboexp as the authors cited in the manuscript, they should also need to compare their work in the context with other ribosome profiling computational deep learning modeling work in the field: just name a few here, such as Tunney R et al (<https://doi.org/10.1038/s41594-018-0080-2>)and RUST (DOI: <https://doi.org/10.1038/ncomms12915>). It would be necessary that the authors to show how their work described in the manuscript stands out, with a detailed side-by-side benchmarking.

2) The current "3-fold cross validation" looks odd, as a common practice is 5-fold cross validation: could the authors explain the rationale?

3) The experimental validation for the in silico mutagenesis and motif analysis, is needed to show the biological validity of the authors' findings, as proposed in our Question 3 to the authors in the 1st round of review. So far, all the claims that the authors put forward are solely based on computational analysis, only interesting suggestions while not yet convincing new biological findings. As the common pitfall of bioinformatics that correlation does not necessarily mean causality, the pinpointed mini-gene experiment on the specific hypothesis would help the readers appreciate the novelty of the authors' work. Particularity, the authors' lab has the established ribosome profiling experimental expertise (the experiments would be reasonably doable/actionable for the authors) and thus the authors should move forward with the experimental validation to convince everyone not only the readers/journal/reviewers, but also themselves. We are not sure the authors' writing in the response that "...perform additional experimental validation..."to be transparent, there is no "experimental validation" in this manuscript.

4) To continue our point in Question 4 of the 1st round of review: On one hand, I appreciate the authors' efforts in improving the writing of their manuscripts to some degree; on the other hand, to help their manuscript, the authors could put together new paragraphs in the manuscript (introduction and discussion) to speak out clearly the underlying logic that they put together in the "response to reviewers" for Questions of reviewer 1 and reviewer 3 (e.g. Q1, Q2 of reviewer1; Q2, Q3 of reviewer3)

Reviewer #3 (Remarks to the Author):

The authors fully addressed all my comments including improving the software, I can now confirm that we have been able to install and run it.

Perhaps it would be useful to add to the readme file a command line for running the test dataset, e.g.

```
cd Riboformer/Riboformer
python3 transfer.py -i ../datasets/GSE139036_disome -m ../models/yeast_disome
```

Pasha Baranov

Reviewer #1 (Remarks to the Author):

The authors have addressed all the concerns that were raised by this reviewer. This reviewer supported the publications in Nature Communications.

We appreciate the reviewer's positive comments on our revised manuscript.

Additional minor comments:

(This reviewer would say that some of the points below should be commented on in the first round of peer review and thus may not be fair to be added in this second round. However, these points may be worth considering to improve this manuscript.)

We would like to thank the reviewer for careful reading of our manuscript.

1. The manuscript defined the window 120 nt or 20 + 20 codons. They are the same but may be helpful to keep consistency throughout the manuscript for easy understanding for readers.

Following the reviewer's suggestion, we have defined the window size in the number of codons throughout the manuscript.

2. On page 6 line 160, it is worth adding the conclusion sentence for the corresponding paragraph.

We have revised the manuscript to include the conclusion sentence for the paragraph.

Revised text (Section "Riboformer accurately clarifies ribosome density"):

Finally, we benchmarked the performance of Riboformer with other deep learning-based models including RiboMIMO and Riboexp and found that it compares favorably (Supplementary Fig. 3, Supplementary Table 4 and Supplementary Note 2). In conclusion, our results highlight the robust performance of Riboformer across different input window sizes, gene expression levels and replicated experiments.

3. On page 7 lines 197-198 "Since translation initiation is..."; this sentence does not sound logical.

We have revised this sentence in the main text to improve clarity.

Revised text (Section "Riboformer allows identification of a limiting step in translation elongation"):

Thus, our results indicate that slow decoding of Trp codons could affect translation elongation, potentially serving as a rate limiting step in protein synthesis.

4. TE should be defined properly (footprints normalized by RNA-seq).

We thank the reviewer for this helpful comment. We have added the definition of TE.

Revised text (Section “Riboformer allows identification of a limiting step in translation elongation”):

We then explored the relationship between translational efficiency (TE) of genes and codon pause scores. Translational efficiency was defined as the ribosome density (RD) normalized by mRNA level as quantified by RNA-seq.

5. On page 7 lines 205-206; “Ribosome stalling in the Tryp codon-rich...”; this sentence and the cited figure may not look match well.

We thank the reviewer for careful reading of our paper and noticing this. We have deleted the figure citation from this sentence.

6. On page 8 line 236, “tRNA binding sites” should be clarified as more concretely (A, P, or E site)

We thank the reviewer for this suggestion. We have specifically mentioned the tRNA binding sites in the revised manuscript. In Fig. 4d, we initially indicated position 0 as the P site, which was a mistake. We have revised the figure to accurately show position 0 as the A site."

Figure R1. Pro codon enrichment scores for all the clusters. This figure is labeled as **Fig. 4d** in the revised manuscript.

7. Fig. 4d was not cited in the manuscript.

We have cited Fig. 4d in the revised manuscript.

Revised text (Section “Riboformer identifies sequence determinants of ribosome collisions”):

Notably, a few clusters have their lowest SIS at the ribosome decoding sites (Fig. 4b, clusters 7-10), indicating that these ribosome collisions are mediated by local sequence features. In these clusters, Pro codons are enriched in all three tRNA binding sites (E, P, and A) (Fig. 4d and 4e), consistent with the well-characterized tendency of Pro residues to slow down translation elongation.

8. For Supplementary Fig. 7a, Cluster 5, ribosome stalling on R-X-K motif was reported previously (DOI: 10.1016/j.celrep.2020.107610, DOI: 10.1016/j.celrep.2016.07.018). These should be cited in the text.

We apologize for the omission of these citations. We have cited the two papers in the revised manuscript. The discovery of the R-X-K motif further demonstrates the practical usefulness of our approach.

Revised text (Section “Riboformer identifies sequence determinants of ribosome collisions”):

Interestingly, the R-X-K motif of cluster 5 is enriched in ribosome collision sites in both humans and zebrafish⁴², and it aligns with the amino acid motifs associated with macrolide-induced ribosome arrest⁴³.

9. On page 9 line 263-264, Pro codon was enriched in A site in Supplementary Fig. 9e-f (for e, 0 position was guessed as A site), but the sentence highlight E and P sites. This point was a contradiction. Similarly, Fig. 4d showed Pro enrichment at +1 site (one codon downstream from A site) as well. This was not reflected in the logo in Fig. 4e.

We thank the reviewer for careful reading of our paper. Pro codons were enriched in A site in Supplementary Fig. 9, and we have revised the main text to correct for this mistake. Additionally, we have revised Fig. 4d, in line with our response to Q6.

Revised text (Section “Riboformer allows interpretation of exacerbated ribosome stalling in aging”):

Further examination of these clusters revealed significant enrichment of Pro codons in the ribosomal A site (Supplementary Fig. 9e-f).

10. On page 10 line 291-293, this reviewer did not agree on the description. First, CHX/TIG was not pre-treated in cells in the paper but rather added in lysis buffer. Second, the difference between long and short footprints was not considered as a “bias”. They simply represent whether A-site was filled with tRNA or not. More precise writing is definitely required.

We have revised our descriptions of this experiment following the reviewer's comments.

Revised text (Section “Discussion”):

We anticipate that our method will also be useful in clarifying ribosome density in eukaryotic samples. While most yeast protocols use cycloheximide to arrest translation in the lysis buffer, Wu et al. found that adding cycloheximide and tigecycline together yields short footprints (~21 nt) from ribosomes with empty A sites and longer footprints (~28 nt) with full A sites⁴⁵. We further demonstrated that Riboformer can be trained to predict these short and long footprint distributions from libraries created with the cycloheximide-only protocol, indicating Riboformer's broad applicability to existing ribosome profiling datasets (Supplementary Table 5).

11. Regarding Fig. 5c, if this reviewer's understanding is correct, the disome peak at position 0 should be "disome", ~-9 position should be trisome (disome + monosome), and ~-18 position should be tetrasome (disome + monosome + monosome). The cartoon inside may not correctly present the data.

Following the reviewer's suggestion, we have deleted the cartoon in Fig. 5c.

12. Supplemental note 1 and 2 were not cited in the manuscript.

We have cited supplementary note 1 and 2 in the revised manuscript.

Reviewer #2 (Remarks to the Author):

We appreciate the authors in taking time to address our concerns in the 1st round of review. Though some improvements, there are several critical issues to be settled before it becomes a manuscript suitable for publication:

We would like to thank the reviewer for the helpful comments.

1) Using deep learning to modeling ribosome profiling is an revisited approach with a few existing methods already in the field: besides the ROSE, RiboMIMO and Riboexp as the authors cited in the manuscript, they should also need to compare their work in the context with other ribosome profiling computational deep learning modeling work in the field: just name a few here, such as Tunney R et al (<https://doi.org/10.1038/s41594-018-0080-2>) and RUST (DOI: <https://doi.org/10.1038/ncomms12915>). It would be necessary that the authors to show how their work described in the manuscript stands out, with a detailed side-by-side benchmarking.

We thank the reviewer for pointing out these references. We have cited the two papers in the revised manuscript. iXnos (Tunney et al) and RUST are two modeling frameworks with diverse applications for ribosome profiling. However, their performance in the specific task of ribosome density prediction is not as good as that of RiboMIMO and Riboexp. These results have been shown in the original studies of RiboMIMO and Riboexp (e.g., **Table 1** in the Riboexp paper and **Table 1** in the RiboMIMO paper). This is the reason why we didn't originally include iXnos and RUST in the comparison. In our work, we have conducted a series of computational experiments to evaluate RiboMIMO, Riboexp and Riboformer across datasets from three species (**Table R1**). Considering these results, we believe that further comparisons with iXnos and RUST would not contribute additional insight to this work.

Table R1. Comparison of prediction performance of Riboformer with that of different baseline methods in terms of the correlation between true and predicted ribosome densities. Riboformer model was evaluated in two modes: "seq only mode" which uses the coding sequence as the only input, and the "full mode" which takes two inputs. We used 10-fold cross-validation tests for Riboformer (n = 10). All models were tested on an identical set of highly expressed genes for each species. The mean \pm SD of the Pearson correlation coefficients and Spearman correlation coefficients are shown. This table is labeled as **Supplementary Table 4** in the revised manuscript.

Dataset (gene numbers)	Metric	Methods			
		Riboexp	RiboMIMO	Riboformer (seq only)	Riboformer (full mode)
E. coli (1005 genes)	Pearson	0.642 \pm 0.005	0.657 \pm 0.024	0.848 \pm 0.011	0.891 \pm 0.002
	Spearman	0.656 \pm 0.002	0.683 \pm 0.013	0.801 \pm 0.015	0.860 \pm 0.003
Yeast (2315 genes)	Pearson	0.587 \pm 0.002	0.581 \pm 0.012	0.886 \pm 0.001	0.932 \pm 0.001
	Spearman	0.489 \pm 0.010	0.491 \pm 0.009	0.798 \pm 0.002	0.869 \pm 0.002

C. elegans (3030 genes)	Pearson	0.511 ± 0.026	0.534 ± 0.009	0.892 ± 0.001	0.920 ± 0.001
	Spearman	0.467 ± 0.010	0.497 ± 0.009	0.784 ± 0.002	0.827 ± 0.002

2) The current “3-fold cross validation” looks odd, as a common practice is 5-fold cross validation: could the authors explain the rationale?

We thank the reviewer for this suggestion. Following the reviewer’s comment, we have used 10-fold cross validation to evaluate the performance of Riboformer. In each fold, one subset of the data was held out as test data; the remaining nine folds were used as training data. Our results are reported in **Table R1** and **Table R2**. We observed little difference between the results of 3-fold and 10-fold cross validation tests in these benchmarking experiments (**Table R3**). Therefore, we continued to use results from 3-fold cross-validation for other computational experiments. We have also revised the supplementary note 1 to clarify this point.

Table R2. Prediction performance of Riboformer in terms of the correlation between true and predicted ribosome densities for all codons in the test dataset. Riboformer model compares ribosome distributions across two datasets and extracts the sequence features driving the difference between them (Fig. 1b). The input and output data for corresponding experiments are included in the table. We performed 10-fold cross-validation tests on four datasets across three species (n = 10). The mean \pm SD of the Pearson correlation coefficients and Spearman correlation coefficients are shown. This table is labeled as **Supplementary Table 1** in the revised manuscript.

Dataset (gene number)	Input data	Output data	Metric	Riboformer
E. coli (1005 genes)	Filtering with the Cm-lysis buffer	Flash-freezing with the high-Mg buffer	Pearson	0.891 ± 0.002
			Spearman	0.860 ± 0.003
Yeast (1608 genes)	Monosome	Disome	Pearson	0.763 ± 0.002
			Spearman	0.620 ± 0.004
Yeast (2315 genes)	0 day (young)	4 days (old)	Pearson	0.932 ± 0.001
			Spearman	0.869 ± 0.002
C. elegans (3030 genes)	1 day (young)	12 days (old)	Pearson	0.920 ± 0.001
			Spearman	0.827 ± 0.002

Table R3. Prediction performance of Riboformer in terms of the correlation between true and predicted ribosome densities for all codons in the test dataset. We compared the results from 10-fold cross-validation tests (n = 10), conducted on four datasets across three species, with the outcomes of 3-fold cross-validation tests (n = 3). The mean \pm SD of the Pearson correlation coefficients and Spearman correlation coefficients are shown.

Dataset	Metric	Riboformer	Riboformer
---------	--------	------------	------------

		(3-fold CV)	(10-fold CV)
E. coli (1005 genes)	Pearson	0.891 ± 0.002	0.891 ± 0.002
	Spearman	0.870 ± 0.001	0.860 ± 0.003
Disome (1608 genes)	Pearson	0.744 ± 0.004	0.763 ± 0.002
	Spearman	0.602 ± 0.006	0.620 ± 0.004
Yeast (2315 genes)	Pearson	0.936 ± 0.006	0.932 ± 0.001
	Spearman	0.862 ± 0.002	0.869 ± 0.002
C. elegans (3030 genes)	Pearson	0.938 ± 0.017	0.920 ± 0.001
	Spearman	0.822 ± 0.002	0.827 ± 0.002

3) The experimental validation for the in silico mutagenesis and motif analysis, is needed to show the biological validity of the authors' findings, as proposed in our Question 3 to the authors in the 1st round of review. So far, all the claims that the authors put forward are solely based on computational analysis, only interesting suggestions while not yet convincing new biological findings. As the common pitfall of bioinformatics that correlation does not necessarily mean causality, the pinpointed mini-gene experiment on the specific hypothesis would help the readers appreciate the novelty of the authors' work. Particularly, the authors' lab has the established ribosome profiling experimental expertise (the experiments would be reasonably doable/actionable for the authors) and thus the authors should move forward with the experimental validation to convince everyone not only the readers/journal/reviewers, but also themselves. We are not sure the authors' writing in the response that "...perform additional experimental validation..." to be transparent, there is no "experimental validation" in this manuscript.

We agree with the reviewer on the difficulty of distinguishing between correlation and causality. However, the objective of this study is to improve the analysis of ribosome profiling datasets from a computational perspective. Riboformer stands in its own ability to model and decipher context-dependent changes of ribosome occupancy, correct the experimental bias, and predict ribosome distribution in new experiments. Further experimental validation is beyond the scope of this study.

The computational capacity of Riboformer aligns with the practical needs of researchers working with ribosome profiling experiments (also see our **response to reviewer 1: Q1 and Q2**, and **reviewer 3's comments** in the first round of revision). The SIS analysis is one of the many applications of Riboformer. Through our analysis, we have demonstrated the impact of proline enriched motifs and polybasic stretches on ribosome stalling. These findings agree with existing knowledge on elongation modulation, but they were previously masked by simple motif analysis. We believe that additional ribosome profiling experiments are not necessary for studying these well-characterized mechanisms.

While our primary focus has been on the methodological development, we acknowledge that further experimental work could expand on our findings. We have provided the following comments to this end in the revised manuscript.

Revised text (Section “Discussion”):

Finally, while our SIS analysis identifies specific ribosome stalling sites that could be mediated by sequence features such as proline-rich motifs, further experimental work will be needed to expand on these findings.

4) To continue our point in Question 4 of the 1st round of review: On one hand, I appreciate the authors’ efforts in improving the writing of their manuscripts to some degree; on the other hand, to help their manuscript, the authors could put together new paragraphs in the manuscript (introduction and discussion) to speak out clearly the underlying logic that they put together in the “response to reviewers” for Questions of reviewer 1 and reviewer 3 (e.g. Q1, Q2 of reviewer1; Q2, Q3 of reviewer3)

We thank the reviewer for this suggestion. We have added a new paragraph in the discussion part that outlines the practical usefulness of Riboformer (Q1, Q2 of reviewer1, reproduced in the box below). We have also dedicated a separate paragraph in the discussion part to address the limitations of Riboformer (Q2, Q3 of reviewer3).

Revised text (Section “Discussion”):

Our framework models the change in ribosome kinetics caused by the experimental protocol, offering a unique opportunity to correct protocol biases in pre-existing datasets and circumvent the need for certain resource-intensive experiments in standard protocols. We have benchmarked its performance by removing experimental artifacts resulting from rapid filtering and the Cm-containing lysis buffer across 16 ribosome profiling datasets produced by four different labs. We anticipate that this method will also be useful in clarifying ribosome density in eukaryotic samples. While most yeast protocols use cycloheximide to arrest translation in the lysis buffer, Wu et al. found that adding cycloheximide and tigecycline together yields short footprints (~21 nt) from ribosomes with empty A sites and longer footprints (~28 nt) with full A sites⁴⁵. We further demonstrated that Riboformer can be trained to predict these short and long footprint distributions from libraries created with the cycloheximide-only protocol, indicating Riboformer's broad applicability to existing ribosome profiling datasets (Supplementary Table 5). Finally, Riboformer can be trained on any pair of ribosome profiling datasets. This flexibility enables *in silico* extrapolation of ribosome densities using a limited number of existing data. Using a trained model to estimate disome profiles based on monosome data, our method can even predict new disome peaks that are not existing in the training datasets.

By simulating the impact of sequence mutations on ribosome occupancy, the Riboformer model identifies the sequences responsible for ribosome collisions, providing insights beyond simple motif analysis. This approach enables a granular classification of ribosome pausing sites, uncovers the impact of amino acid charges and mRNA structure on ribosome collisions and identifies the effect of proline enriched motifs on ribosome stalling in young and aged yeast...

Reviewer #3 (Remarks to the Author):

The authors fully addressed all my comments including improving the software, I can now confirm that we have been able to install and run it.

Perhaps it would be useful to add to the readme file a command line for running the test dataset, e.g.

```
cd Riboformer/Riboformer
```

```
python3 transfer.py -i ../datasets/GSE139036_disome -m ../models/yeast_disome
```

We thank the reviewer for the positive comments on our revised manuscript. We have added these commands in the updated readme file (<https://github.com/lingxusb/Riboformer#applying-trained-riboformer-model-on-new-dataset>).

Pasha Baranov